# Are you confident enough to act? Individual differences in action control are associated with post-decisional metacognitive bias

**Wojciech Zajkowski**[1,2]\*, **Maksymilian Bielecki**[1], **Magdalena Marszał-Wiśniewska**[1]

1 Department of Psychology, SWPS University of Social Sciences and Humanities, Warsaw, Poland,
2 Cardiff University Brain Research Imaging Centre, School of Psychology, Cardiff University, Cardiff, United Kingdom

\* wojciech.zajkowski@riken.jp

**Data Availability Statement:** Data, together with reproducible code for analyses and figures can be found in an online OSF repository under the URL: https://osf.io/6dfs3/.

## Abstract

The art of making good choices and being consistent in executing them is essential for having a successful and fulfilling life. Individual differences in *action control* are believed to have a crucial impact on *how* we make choices and *whether* we put them in action. *Action-oriented* people are more decisive, flexible and likely to implement their intentions in the face of adversity. In contrast, *state-oriented* people often struggle to commit to their choices and end up second-guessing themselves. Here, we employ a model-based computational approach to study the underlying cognitive differences between action and state-oriented people in simple binary-choice decision tasks. In Experiment 1 we show that there is little-to-no evidence that the two groups differ in terms of decision-related parameters and strong evidence for differences in metacognitive bias. Action-oriented people exhibit greater confidence in the correctness of their choices as well as slightly elevated judgement sensitivity, although no differences in performance are present. In Experiment 2 we replicate this effect and show that the confidence gap generalizes to value-based decisions, widens as a function of difficulty and is independent of deliberation interval. Furthermore, allowing more time for confidence deliberation indicated that state-oriented people focus more strongly on external features of choice. We propose that a positive confidence bias, coupled with appropriate metacognitive sensitivity, might be crucial for the successful realization of intentions in many real-life situations. More generally, our study provides an example of how modelling latent cognitive processes can bring meaningful insight into the study of individual differences.

## Introduction

We make a plethora of decisions every day, ranging from trivial ones such as choosing what to have for lunch to life-changing choices about our future. One of the main inter-individual factors believed to influence the effectiveness of one's decision-making and execution skills is the level of *action control*.

**Funding:** Preparation of this article was supported by Polish Ministry of Science and Higher Education Grant 'Diamentowy Grant' (DI2014 0081 44). The first author (WZ) was the only benefitiary of the grant. Information about the grant can be found under this URL: https://www.gov.pl/web/edukacja-i-nauka/diamentowy-grant The funders had no role in study design, data collection and analysis, decision to publish, or preparation of the manuscript.

**Competing interests:** The authors have declared that no competing interests exist.

Action control is a construct introduced by Kuhl [1–3], which describes individual differences in volitional abilities concerned with initiating, maintaining, and finalizing planned actions. Measured by the Action Control Scale (ACS-90) [4], it is a unique construct related to differences in cognitive control (moderate positive correlations with self-consciousness and self-monitoring), as well as emotional processing (moderate negative correlations with negative thoughts, emotions, and affect or reactivity to stress) [5]. Diefendorff and colleagues [6] showed that only a small portion of variance in ACS scores (between 6% and 28% depending on the subscale) can be accounted by a combination of Big Five [7] personality factors. Its distinctive feature is the focus on practical behavioural consequences—the ability to maintain and enact intentions [2]. The construct is built around this central premise, claiming that the abilities to downregulate negative and upregulate positive affect act as a specific cognitive-control mechanism mediating the behavioural outcomes [8]. People with a high level of action control (referred to as *action-oriented*) are more flexible, adjustable, and well-versed in realising their goals, especially under challenging conditions such as time pressure, stress, or low probability of success. Low levels of action control (*state-orientation*) are associated with opposite effects, described as *volitional inhibition*. When faced with adversity, state-oriented people are not capable of adjusting their plans, and more likely to abandon their commitments [9].

Action control exerts influence on one's life satisfaction and success. It is associated with greater well-being [10], higher satisfaction of self-regulatory abilities in the work environment [11], greater work achievements and recognition [6, 12, 13], and better academic performance [14]. State-oriented people, on the other hand, have less satisfying relationships [15], display unhealthy habits [16–18], and exhibit a greater risk of developing depression [19, 20]. How can differences in action control alter our wellbeing to such an extent? We aim to answer this question by focusing on the two core aspects of decision making: the cognitive mechanisms responsible for generating choices and post-decisional processing, *i.e.* our perception of the selected course of action.

## Metacognition and action control

The concept of action control is interesting not only by itself, but also from a broader perspective of metacognition. The critical period between decision and action is accompanied by metacognitive processes facilitating commitment. These include reducing cognitive dissonance [21] and intention shielding [22], which allow us to feel more confident about our choices and follow through with realising them [23]. The opposite—a lack of attitude adjustment or weak shielding of intended action could result in hesitation, reluctance, and frequent decision shifts. These processes can be captured by measuring post-decisional confidence—an area of cognitive research that has been very fruitful in recent years [24, 25]. Regarding confidence judgements, we use the term <u>post-decisional</u> instead of more frequently encountered term *retrospective* [26] to accentuate the role of choice in shaping the metacognitive process.

Methodological advances in metacognition research enabled dissecting metacognitive *accuracy* (or skill) from *bias* by utilizing the signal detection theory framework [27, 28]. Bias is defined as the absolute difference in confidence from a given reference point. Stated simply, a positive bias would imply overconfidence, given an arbitrary *status quo*. Recent studies on individual differences show that increased confidence bias is associated with high trait optimism [29] as well as compulsive behaviour cluster [30]. Negative bias (relative underconfidence) has been linked to symptoms of anxiety and depression [30]. Metacognitive accuracy is the sensitivity of confidence judgements (sometimes referred to as metacognitive sensitivity [28]). More accurate judgements mean a better insight into the decision process (higher discriminability). A highly accurate agent would have relatively low confidence in their

incorrect choices and relatively high in correct ones. Accuracy declines with ageing [31] and is consistently subpar among people holding radical beliefs [31].

The association between bias and accuracy is complex [27, 28, 32, 33]. A highly biased agent can still be very accurate in his judgements by having a very high or low mean confidence score (reflecting over or underconfidence) but discriminating well between correct and incorrect choices. Conversely, a well-calibrated agent can be terrible at distinguishing between correct and incorrect choices. It seems likely that action control has a critical influence on *how* and *what* we think about our choices after initial commitment. Indeed, state-orientation has been shown to reduce alternative spreading, a phenomenon where alternative value grows after being chosen and shrinks after being rejected [34]—one of the hallmarks of cognitive dissonance reduction [35]. It is easy to imagine how a perpetual lack of confidence could lead to a spiral of self-doubt inhibiting the realisation of meaningful plans.

## Action control and the stages of decision making

A difference in action control might be also associated with decision-making process. Dibbelt and Kuhl [36] propose a 4-stage model of decision making which include: *1) information selection*, *2) information integration*, *3) termination* and *4) postdecisional maintenance*. The authors postulate that differences in action control influence how people approach each of the stages. Specifically, action-oriented people facilitate selection by simplifying the choice space (stage 1), integrate information in a more parsimonious fashion (stage 2), utilize a more efficient and flexible decision rule (stage 3), and maintain the decision-related intention longer (stage 4). Despite the heuristic potential of the model postulated by Dibbelt and Kuhl, few studies have tried to tackle these theoretical assumptions directly. Available empirical findings show differences between the groups only in extraordinary and stressful circumstances such as high demand [37, 38], high conflict [39], high stakes, or time pressure [40]. A lack of significant differences between groups in non-stressful control conditions suggests that the relation between decision making and action control might be more complex than primarily assumed by Dibbelt and Kuhl. Accordingly, some authors claim that state-oriented individuals might function equally well or even better under low pressure than their action-oriented counterparts [9].

## The present study

This study aims to characterize the differences among people with high and low action control by systematically comparing both decision-related and meta-cognitive aspects of their behavior. We employ a set of 2-alternative perceptual and preference-based choice tasks, manipulating prior information, decision urgency and choice difficulty (Experiment 1), decision type, and the interval between choice and confidence judgement (Experiment 2). In addition to raw measures of performance (accuracies, reaction times, confidence judgements), we also compare latent variables of interest, such as the speed of evidence accumulation and decision caution (as defined by Drift Diffusion Model) [42], as well as pre-decision bias, and metacognitive accuracy (based on signal detection theory framework) [27]. These measures are of interest since they correspond directly to the four stages of the decision-making process described by Dibbelt and Kuhl [36]. A simplifying strategy for information selection (1) in a two-alternative choice can be achieved in one of two ways. First, a strong initial prior bias towards one of the options can facilitate selection, especially in cases where sensory information is ambiguous. Additionally, one might display a bias when observing the incoming evidence by weighting more heavily evidence in favour of the option predicted to be the better of the two. Both of these mechanisms can be tested directly by manipulating the prior probability of the outcome.

**Table 1. Model-based parameters of interest derived from the Drift Diffusion Model [42] and meta-d' [27].**

| Parameter name | Description | Cognitive interpretation |
|---|---|---|
| Drift rate $v$ (DDM) | Information uptake per unit of time | High values indicate faster processing. |
| Decision threshold $a$ (DDM) | Response criterion | High values indicate more cautious response strategy. |
| Accumulation starting point $z$ (DDM) | Asymmetries in the response criterion | Values further from the midpoint suggest initial bias towards one of the alternatives. |
| Metacognitive accuracy $meta\text{-}d'$ | sensory evidence available for metacognition in signal-to-noise ratio units [27] | High values indicate greater metacognitive sensitivity. |

The efficiency of information integration (2) can be expressed as the speed of evidence accumulation. Termination speed and flexibility (3) can be measured by manipulating choice speed vs accuracy tradeoff and estimating decision thresholds. Finally, decision maintenance (4) can be related to post-decisional confidence judgements, which reflect the certainty associated with the decision and influence the likelihood of implementation, as predicted by Kuhl's model [36]. Confidence judgements can be broken down into two components: magnitude and accuracy (sensitivity). Table 1 describes model-based parameters of interest. Table 2 reflects theoretically-driven hypotheses regarding primary (decision-related) and secondary task (post-decision confidence) performance. Following action-driven theories of dissonance [41], we expect differences in post-decisional confidence magnitude to be most meaningful. In our study, we stay agnostic with regard to finding group differences in the decision-making phase. While predicted by the aforementioned theoretical framework, previous research suggests that extraordinary circumstances such as heightened stress or mental pressure might be necessary to induce an effect [9, 37–40]. We also do not make specific predictions regarding differences in metacognitive accuracy since neither theory nor empirical findings justify a prediction. On the one hand there is a reason to believe that confirmatory post-decisional biases distort accuracy in favour of making sure the action is accomplished [23]. According to this view, we expect state-oriented people who are less affected by these biases to exhibit higher metacognitive sensitivity. On the other hand, higher action control might be associated with greater insight into the decision-making process, improving metacognitive sensitivity. More detailed justification of the proposed set of hypotheses will be presented in the context of each of the studies.

## Ethics approval

The current study received ethical approval from the SWPS University Research Ethics Committee (opinion no. 9/2015).

## Software and data accessibility

Data and reproducible R (https://cran.r-project.org/) code for analyses and figures can be found in an online repository (https://osf.io/6dfs3/).

**Table 2. Parametrization of tested hypotheses based on the model of decision-making process by Dibbelt and Kuhl [36] and operationalized using Drift Diffusion Model [42] and meta-d' [27].**

| Decision Phase | Cognitive processes | Model parameters | Testable predictions |
|---|---|---|---|
| (1) Information Selection | Pre-decision bias | Starting point $z$<br>Drift rate $v$ | $z$: action > state<br>$v$: action > state |
| (2) Information Integration | Speed of evidence accumulation | Drift rate $v$ | $v$: action > state |
| (3) Termination | Stopping rule | Threshold $a$<br>Difference in threshold $\Delta a$ | $a$: action < state<br>$\Delta a$: action > state |
| (4) Postdecisional maintenance | Metacognitive bias and accuracy | Confidence bias; meta-d' accuracy | bias: state < action |

## Experiment 1

Existing studies of action control and decision-making show differences between groups only in the presence of a strong affect [9]. Hence, it is difficult to distinguish the effects of basic cognitive elements of the decision process, such as information integration or termination speed, from more complex interactions driven by the stressors. To disentangle this relation, we use a design that manipulates potential latent decision variables of interest without affective manipulations. Our participants performed two classic 2-alternative forced-choice (2AFC) perceptual decision paradigms: a *Random-Dot Motion* (RDM) task and a *Face-House Discrimination* (FHD) task. In RDM, we manipulate decision caution by instructing participants to focus on their choices' speed or accuracy. In FHD, we manipulate pre-decision bias by providing *congruent*, *incongruent* or *neutral* cues before each choice. Additionally, we control for choice difficulty between tasks (RDM being relatively difficult while FHD relatively easy). In both tasks, participants rate their decision confidence after each choice.

We utilize *Drift Diffusion Model* (DDM; [42]) to estimate evidence accumulation speed, pre-decision bias and decision threshold, as well as *meta-d'* model [27, 43] to measure metacognitive accuracy. DDM is one of the most widely used among the family of sequential sampling models [44]. Such models assume that decision-relevant evidence is sampled sequentially and accumulated until it crosses a predefined decision threshold. This dynamic feature of the model sets it apart from more frequently encountered models in personality psychology [45], such as process dissociation [46] or signal detection [47], as it allows to incorporate both accuracy and decision time into a single model.

DDM assumes that this decision process is driven by noisy evidence accumulation that can be characterized by *accumulation speed* (*v*, also called *drift-rate*), *decision threshold* (*a*) and a *starting point* (*z*), located somewhere between the lower and upper decision threshold. Drift-rate is an indicator of the quality of evidence processing. Depending on the response coding scheme, higher values can indicate either better performance (*accuracy coding*) or a larger bias towards one response category (*stimulus coding*). Threshold relates to decision caution with greater values, indicating that more information is needed before making a choice. Starting point relates to pre-choice bias. When its location is equidistant from both bounds (choice options), the amount of evidence necessary for either option is equal (no pre-choice bias). If it is closer to one of the bounds, less evidence is needed to commit to the option associated with that bound (*e.g.*, when a person needs little evidence to believe in something consistent with her pre-existing beliefs but requires way more convincing to believe in something inconsistent). Other non-decisional processes such as stimulus encoding time and motor execution are captured by *non-decision time* parameter *Ter*. DDM studies have been very fruitful for studying individual differences [48] (for a more in-depth introduction to DDM in the realms of social and personality psychology, see also: [45]).

Meta-d' model estimates metacognitive accuracy using signal detection theory framework [27]. The basic idea is to compare the overlap between confidence rating distributions following correct and incorrect choices. A high overlap would indicate that correct choices are rated similarly to incorrect ones, indicating poor judgement. Conversely, low overlap in the expected direction indicates high discriminability. One of the crucial advantages of meta-d' over similar measures of metacognitive sensitivity is that it is bias-free (for a more technical argument, see: [27, 28]).

We parametrize tested hypotheses in terms of differences in underlying latent variables of interest manifested by model-derived parameters.

***Hypothesis 1.*** Differences in information selection. Biased information selection would be reflected by preferentially sampling only information consistent with a given prior

hypothesis. We test this by providing congruent, incongruent or neutral cues before each choice. Bias can act on two DDM parameters: sampling starting point $z$ (reflecting bias in evidence needed for choice commitment), as well as drift-rate $v$ (reflecting bias in the sampling process; [49]).

*Hypothesis 2.* Differences in information integration. Information integration can be expressed as the speed of evidence integration. Faster integration would lead to faster and more accurate responses. It depends on a person's skill and evidence quality (manipulated by task difficulty). This process is reflected by DDM drift parameter $v$.

*Hypothesis 3.* Differences in termination. Termination is the sampling stopping rule or threshold. A high threshold indicates that an agent requires more information to commit to a choice. Conversely, a flexible agent would adjust the threshold depending on the task demands. We control this parameter by instructing participants to focus on either decision accuracy or speed. This process is reflected by DDM threshold parameter $a$. Here, in addition to testing the difference in an absolute threshold, we can also estimate the flexibility of threshold adjustment dependent on task instruction. High flexibility would be associated with a large difference in threshold ($\Delta a$) between accuracy and speed conditions ($a_{\text{accuracy}} - a_{\text{speed}}$).

*Hypothesis 4.* Differences in post-decisional maintenance. Post-decisional maintenance refers to metacognitive confidence judgements. Meta-d' parameter reflects metacognitive accuracy, and the confidence magnitude indicates bias.

## Method

**Participants.** 724 participants from SWPS University of Social Sciences and Humanities completed the Action Control Scale (ACS-90) [4] (Polish adaptation by Marszał-Wiśniewska [50]). Written informed consent was obtained before filling the questionnaire from all participants. We calculated action control as a combined score on two subscales of ACS-90: *Hesitation* and *Preoccupation* (total of 24 yes/no questions) and selected 60 participants with extreme scores for each Experiment (2 groups of high and low scoring participants). We used the combined scales approach since our hypotheses were independent of any specific dimension of action control, prioritizing recruiting participants with the most extreme scores. Additionally, we ensured that all selected participants scored below or above the median on both subscales. Since the recruitment was performed on a large scale, not all questionnaire takers were willing to participate in the main experiment. Overall, we faced about ~50% rejection rate. The final participant pool consisted of 30 action-oriented participants (14 female and 16 male, $M_{\text{age}}$ = 23.43, $SD_{\text{age}}$ = 4.45; $M_{\text{ACS score}}$ = 18.5; $SD_{\text{ACS score}}$ = 2.08) and 30 state-oriented participants (19 female and 11 male, $M_{\text{age}}$ = 21.67, $SD_{\text{age}}$ = 3.35; $M_{\text{ACS score}}$ = 3.36, $SD_{\text{ACS score}}$ = 1.63). When compared against the distribution of all ACS scores obtained in the screening phase, the means in two contrast groups were separated by 2.81 SD. Four state-oriented participants did not show up for the second experimental session, making the number of state-oriented individuals equal to 28 for each of the tasks (since the tasks were counterbalanced across sessions).

**Action Control Scale (ACS-90).** ACS-90 scale [4] consists of 3 subscales, each containing 12 questions describing situations. Participants choose the answers that best describe them. The hesitation subscale relates to the ability to initiate intended actions. Sample item: *When I'm getting ready to tackle a difficult problem*: A) *It feels like I am facing a big mountain that I don't think I can climb* (B) *I look for a way that the problem can be approached in a suitable manner. The* Preoccupation subscale is related to the ability to implement action in the face of adversity. Sample item: *When I am told that my work has been completely unsatisfactory*: A) I

*don't let it bother me for too long. B) I feel paralyzed. The* volatility subscale relates to the stability of action execution. Sample item: *When I am busy working on an interesting project*: *A) I need to take frequent breaks and work on other projects B) I can keep working on the same project for a long time*. Despite the naming of the subscales relating to the state-oriented spectrum, high scores indicate high action control. In both experiments, we only considered scores on the Hesitation and Preoccupation subscales, excluding the Volatility dimension, in agreement with Kuhl's idea that the ability to escape state-oriented control (as described by Hesitation and Preoccupation) is independent of the ability to remain in the action-oriented state (Volatility) [6]. Consistent with this idea, empirical findings show high correlations between the two action initiation dimensions and a close to zero association with Volatility [2, 4].

**RDM task.** In the random-dot motion task, participants observed a cloud of white dots on a black background and decided whether the majority of them moved to the right or left side. The motion stimuli were similar to ones used previously [51, 52]. Each decision was followed by a confidence judgement on a 6-point scale ranging from 50% (random guess) to 100% (absolute certainty) in 10% increments. The task consisted of 3 phases: initial training, difficulty calibration block and the main task (3 blocks). To match the difficulty level across participants, each subject completed a calibration block of 200 trials of randomly interleaved stimuli with varying motion strength (10, 20, 40 or 80% coherence levels). Following the procedure introduced by Palmer, Huk and Shadlen [52], we fitted the proportional-rate diffusion model to the mean response times and accuracy data of this block and interpolated the motion strength for 65% accuracy from the psychometric curve. We used this individually assessed coherence level throughout the main task. The main experiment consisted of 3 blocks of 54 trials. *Speed* versus *accuracy* emphasis was manipulated across trials. A single trial consisted of: 1) fixation cross (500 to 1000 ms, uniformly distributed), 2) emphasis manipulation screen with big letters 'SZ!' indicating response speed or 'PREC' indicating accuracy emphasis, 3) stimulus presentation (choice period; maximum of 1500 ms), 4) confidence rating screen. The task was programmed using Psychophysics Toolbox Version 3.0.8 [53].

**FHD task.** In the face-house discrimination task, participants discriminated between noise-degraded stimuli representing the two categories. Each choice was preceded by a cue providing a probability of observing a given stimulus (80% House, 80% Face or 50%) and followed by a confidence judgement on 6-point scale ranging from 50% (random guess) to 100% (absolute certainty) in 10% increments. The stimuli consisted of 42 neutral-expression faces and 42 house grayscale images, used in a previous study by Dunovan and colleagues [54]. The task consisted of 5 blocks of 96 trials. Given the cue, the following stimulus could be congruent (53.33% of trials), incongruent (13.33% of trials) or neutral (33.33% of trials). Each trial consisted of 1) fixation cross (500-1000ms), 2) cue presentation (500ms), 3) stimulus presentation (choice period, maximum of 1250 ms), 4) confidence rating. The task was programmed using *PsychoPy* Python library.

**Procedure.** When invited to participate in the study subjects were informed that they qualified 'based on questionnaire results'. In order to avoid influencing performance, no further detail was provided. Participants performed the experiment in two sessions. Each dedicated to one of the tasks (Fig 1). The task order was counterbalanced across participants. Both sessions were completed within a two-week period. In both tasks, participants responded using 'z' and 'm' keys to indicate binary choice and number keys '1' to '6' when rating confidence. Participants were instructed that the confidence scale maps linearly to the subjective probability of being correct, such that *1* equals *50%* (or lower), *2* equals *60%*, up to 6 (~100%). In the RDM task, 'z' corresponded to '*left*' and 'm' to '*right*' response; in FHD task, the key-response bindings were counterbalanced between participants so that for half of the participants '*z*' represented faces and '*m*' houses and *vice versa*. At the end of the study, participants

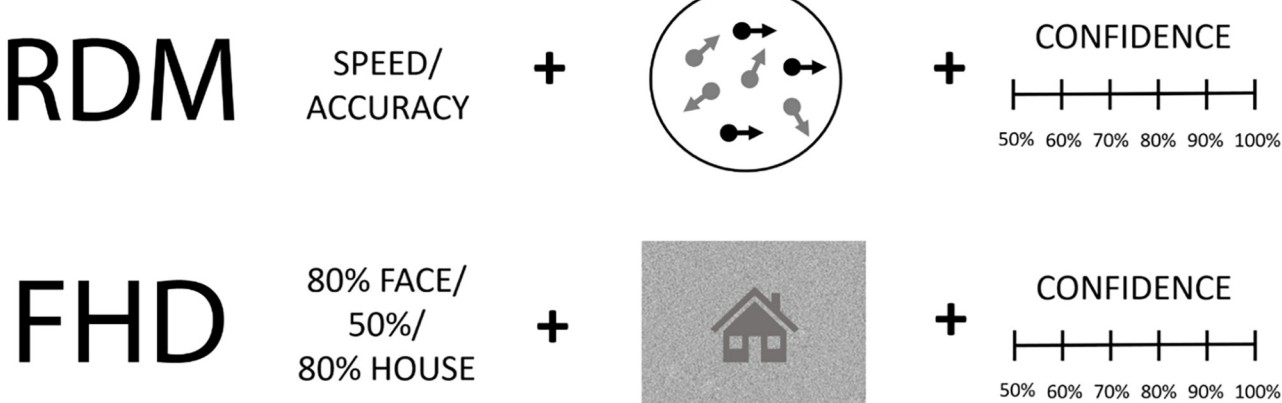

**Fig 1. Experiment 1 task design.** Upper row: random-dot motion (RDM) task. Lower row: face/house discrimination (FHD) task.

were thanked for their participation, rewarded with course credit (if applicable), debriefed, and told why they were selected for the experiment. Each experimental session lasted between 30 to 60 minutes. All tasks were performed offline and took place in a testing lab on the university's premises.

**Measures.** For each task, we analyse a series of variables of interest and compare them across groups using mixed-design *ANOVA* (where applicable; for an exception, see DDM section). We distinguish between *directly observable* measures of performance (RT, accuracy and confidence magnitude) and *latent*, model-derived measures. These include DDM parameters (threshold, threshold difference, starting point bias and drift rate; see Experiment 1 introduction), processing efficiency scores [55, 56] and metacognitive accuracy scores. Across the paper, if the *ANOVA* sphericity assumption is violated, we report results and p-values calculated with Greenhouse-Geisser correction.

*DDM*. To estimate DDM parameters, we employ hierarchical drift diffusion model (*HDDM*) [55]. A hierarchical Bayesian estimation method is particularly well suited for samples with relatively smaller trial counts [57, 58]. Because of large model space (assuming 3 parameters of interest that can vary as a function of any combination of 3 experimental factors results in 64 candidate models for RDM task and 128 for FHD), we simplify the search by performing a two-step model fitting routine. We first search for the best fitting model assuming no between-group differences and only testing the influence of within-participant manipulated factors (8 candidate models in RDM; 16 in FHD). This gives us a good indication of whether the best model accurately describes the data and which of the parameters reliably vary between conditions. In stage 2, we fix the within-person model structure to the one represented by the winning stage 1 model (now referred to as *no-trait model*) and fit models that vary the influence of action control on model parameters (7 additional models in each task). We then compare the between-person models to the no-trait model to see whether including *action control* improves model fit.

Posterior distributions are generated using 5000 samples drawn using the *Markov Chain Monte Carlo* method [59] with a 1000 sample burn-in. Weakly informative prior parameter values were based on a collection of 23 studies reporting best fitting DDM parameters in decision-making tasks [55]. We compare the models using *Deviance Information Criterion* (DIC) [60], a metric of fit that considers model complexity, where lower scores indicate better fit. Since DIC can be a biased and imperfect measure [55], we adopt a conservative threshold proven as reliable in other works using the HDDM toolbox. Conventionally, differences larger

than 10 in DIC scores are regarded as significant [61]. While this assumption msight be considered an oversimplification, it has proven to work well and has been adapted in many empirical works using the HDDM toolbox [54, 62, 63]. We assess model convergence by running five chains of the model and calculating the Gelman-Rubin convergence metric *R-hat* [64]. *R-rat* measures the ratio of between to within chain variance. Measures below 1.1 signify good convergence [65]. To see if the model can accurately account for the real data, we run a posterior predictive check [55] by simulating data generated from the model and checking if it can reproduce the quantitative and qualitative patterns present in the real dataset.

Among other advantages, hierarchical Bayesian estimation results in *shrinkage* [66], making individual estimates influenced by the group distribution and *vice-versa*. This property is desirable, leading to more accurate predictions [66, 67]. Since shrinkage makes individual parameter values dependent on the group distribution, a classical ANOVA analysis is inadvisable. Therefore, we draw inferences by comparing the posterior distribution of parameter values. For example, if parameter *v* is larger in condition X for 90% of posterior samples compared to condition Y, we have 90% certainty that v in condition X is, in fact, larger (for a similar approach, see: Cavanagh et al. [68]). Analogous to the frequentist account, we infer a difference between groups to be significant if 95% or more of the sample values are larger in one of the two compared posterior distributions. To distinguish these probabilities from their frequentist counterparts, we use the convention of denoting the probability with a capital letter *P* and providing the complement of the frequentist probability. For example, a 97% difference in posterior distributions would be denoted as $P = .97$.

*Efficiency scores.* Sometimes, measures of RT and accuracy alone do not provide a complete picture of performance efficiency. For example, if one group is slightly slower and slightly less accurate, differences between these measures might not be significant, but a measure combining the two could be sensitive enough to reveal one group being more efficient. To take that into account, we use another joint measure of RT and accuracy—*linear integrated speed-accuracy score* (LISAS) [55]. The formula for calculating the coefficient is to add the mean RT for correct responses in a given condition to the overall error probability multiplied by the ratio of the standard deviations of these two variables. We chose LISAS among a set of similar methods since it is the most sensitive and least biased estimator of processing efficiency when compared to similar measures [55, 69]. This parameter has a similar interpretation to *drift-rate*. A the same time, it has the advantage of being more easily derived and estimated, serving as a complementary measure in our analyses.

*Metacognitive accuracy.* To calculate metacognitive accuracy scores (meta-d'), we used *HMeta toolbox* [43] using the maximum likelihood method for obtaining single-subject parameter estimates.

## Results

**ACS-90 scale internal consistencies.**   A reliability analysis was carried out for ACS-90 subscales within the sample. Cronbach's alpha showed a reliability of $\alpha = 0.79$ for the Hesitation subscale, $\alpha = 0.79$ for the Preoccupation subscale, and $\alpha = 0.85$ for the combined subscales. Our sample results are closely related to the original scale consistencies (Hesitation $\alpha = 0.78$, Preoccupation $\alpha = 0.70$, combined $\alpha = 0.81$) [4]. The correction between the two subscales was $r = 0.49$, similar to the ACS-90 correlation of $r = 0.43$ [4] as well as the Polish adaptation ($r = 0.45$) [50].

**RDM task.   *Data preprocessing*.** We removed response times shorter than 200 ms and outliers over 3 standard deviations from the mean per subject and condition (less than 2% of trials). No participant was excluded since all performed above chance level.

**Coherence.**   We compared dot coherence levels across groups to account for the possibility that one of the groups might be more proficient in the task, which could be masked by the initial difficulty adjusting procedure. A two-sided t-test found no difference between the groups $t(40.499) = 1.241$, $p = .22$.

**RT and accuracy.**   For each variable of interest (RT, accuracy), a mixed-design ANOVA was performed using condition (accuracy, speed) and motion direction (left, right) as within-person factors, and action control (action, state) as a between-person factor. Reaction times in the *speed* condition ($M = 645$ ms) were faster than in *accuracy* ($M = 750$ ms), showing our manipulation was successful, $F(1,56) = 86.86$, $p < .001$, $\eta_g^2 = .12$. Accuracy was higher in *accuracy* ($M = 67.0\%$) compared to *speed* condition ($M = 65.4\%$), however the difference did not reach significance $F(1,56) = 2.44$, $p = .12$. This effect might be caused by the high difficulty of the task and an *event-related* design, where conditions changed randomly from trial to trial, making the task significantly more challenging and requiring constant switching. No between-groups differences reached significance.

**Confidence magnitude.**   A 2 x 2 x 2 mixed-design ANOVA revealed main effects of condition and action control. Confidence was higher in accuracy ($M = 86.2\%$) vs. speed condition ($M = 83.3\%$), $F(1,56) = 38.69$, $p < .001$, $\eta_g^2 = .04$. Action-oriented individuals were more confident ($M = 87.8\%$) than state-oriented ones ($M = 81.6\%$), $F(1,56) = 13.99$, $p < .001$, $\eta_g^2 = .18$. No other effects reached significance.

**DDM.**   In the first stage, we compared 8 models, varying the influence of choice emphasis (speed or accuracy) on 3 parameters: drift-rate *v*, decision threshold *a*, and non-decision time *Ter*. Responses were represented using an accuracy coding scheme, where one bound of the diffusion process represents correct responses and the other incorrect ones. Condition-dependent threshold changes would indicate strategic adjustments in the amount of information necessary for decision commitment; for example, lowering the threshold when speed is emphasized, consistent with the theoretical prediction of optimal behaviour [69] and empirical findings [70]. Condition-dependent differences in drift rate *v* would indicate that the speed cue mobilizes participants to process the information faster, a finding most commonly observed in participants new to the task [64]. Finally, differences in non-decision time would indicate that the cue also influences motor preparation or execution times [64, 71].

The best fitting model (DIC score of 8445, a 23-point difference compared to the second best) had separate *a* and *Ter* distributions for each condition and a single *v*. All parameters displayed excellent convergence (*R hat*<1.01). A posterior predictive check analysis [55] validated the adequacy of the model, showing that it could accurately reproduce quantile distributions of RT and the accuracy data. A comparison of posterior group parameters revealed that threshold was larger in the accuracy condition $M_{ACC} = 412$ ms; $M_{SP} = 324$ ms, $P = .945$. Non-decision times were longer in accuracy emphasis condition $M_{ACC} = 412$ ms; $M_{SP} = 324$ ms, $P > .999$. Single group-level drift rate suggests that there is no reason to believe that evidence accumulation speed was dependent on trial instructions. These results suggest that apart from the expected strategic adjustment of the threshold, our manipulation also influenced the speed of motor execution accounted for by non-decision time, a finding also reported in previous studies [64, 71].

In the second stage, we compared the winning stage 1 model (now referred to as *no-trait model*) with 7 additional models, varying the effects of action control on *a*, *v*, and *Ter*, and keeping the within-effect structure consistent with the no-trait model. All the models displayed very similar *DIC* scores (between 8441 and 8448), indicating that adding action control as a between-person factor did not significantly improve model fit. Therefore, we found no evidence of differences in decision-relating processing between the groups.

**Efficiency.** A 2 x 2 mixed-design ANOVA (condition x action orientation) of LISAS scores revealed a significant effect of condition on processing efficiency. Efficiency was significantly better in the *speed* condition, $F(1,56) = 83.64$, $p < .001$, $\eta_g^2 = .09$. This is unsurprising since faster responses in the *speed* condition were not associated with significantly reduced accuracy. No evidence that efficiency varied across groups was found.

**Metacognitive accuracy.** 2 x 2 mixed-design ANOVA of meta-d' scores showed significant effects of condition and action control on metacognitive accuracy. Metacognitive sensitivity was higher in speed ($M = 1.41$) compared to accuracy condition ($M = 0.88$), $F(1,56) = 4.37$, $p = .04$, $\eta_g^2 = .04$. Action-oriented individuals showed higher metacognitive accuracy ($M = 1.39$) than state-oriented ones ($M = 0.87$) $F(1,56) = 4.74$, $p = .03$, $\eta_g^2 = .04$.

**FHD task. *Data pre-processing.*** We removed response times shorter than 200 ms and outliers over 3 standard deviations from the mean per subject and condition (less than 2% of trials). After initial data screening, one participant (*action-oriented*), was removed, since he responded according to the cue on 100% of the trials and performed randomly (47.1% accuracy) in the neutral condition, indicating he did not draw any information from the stimuli.

**RT and accuracy.** We analysed the variables of interest as a function of stimulus type (*face*, *house*) and cue type (*congruent*, *incongruent*, *neutral*) as within person, and action control (*action*, *state*) as between person factors. RTs were faster for faces ($M = 850$ ms) compared to houses ($M = 928$ ms), $F(1, 56) = 214.94$, $p < .001$, $\eta_g^2 = .18$. Cue type also affected RTs, $F(2,110) = 7.74$, $p < .01$, $\eta_g^2 = .01$. Post-hoc contrasts revealed that effect was driven by congruent trials ($M = 880$ms) being significantly faster than incongruent ($M = 897$ms), $t(110) = 3.907$, $p < .001$, while no-cue trials ($M = 889$ ms) did not significantly differ from the rest. No group effects reached significance. Accuracy was modulated by cue type, $F(2,110) = 13.32$, $p < .001$, $\eta_g^2 = .07$, with congruent ($M = 91.7\%$) and neutral trials ($M = 88.8\%$) being both more accurate than incongruent ($M = 86.2\%$). Main effects of cue type on accuracy and *RT* indicate that the experimental manipulation was successful. A significant interaction of cue type and *stimulus*, $F(1.51, 83) = 14.05$, $p < .001$, $\eta_g^2 = .01$, showed that the influence of cue type was stronger if the stimulus was a house. Together with slower RTs for house stimuli, as well as previous findings indicating lower discriminability of houses compared to faces [54, 62] this suggests that participants adjusted their strategy by relying on the house cue to a greater extent, since the stimulus evidence was less reliable. No group effects reached significance.

**Confidence.** 2 x 2 mixed-design ANOVA indicated confidence was higher when a face was present ($M = 92.1\%$) compared to a house ($M = 87.4\%$), $F(1,55) = 55.97$, $p < .001$, $\eta_g^2 = .10$, and when the cue was congruent ($M = 90.7\%$) or neutral ($M = 90.0\%$) compared to incongruent ($M = 88.6\%$), $F(2.13, 117.57) = 12.97$, $p < .001$, $\eta_g^2 = .01$. Similarly to accuracy, the interaction between cue and stimulus, $F(1.77, 97.32) = 22.21$, $p < .001$, $\eta_g^2 = .01$, indicated the cue influencing confidence to a higher degree when a house was presented. Lower confidence when houses were presented coupled with a higher reliance on *house* cue is consistent with houses being in general more difficult to distinguish, leading to higher uncertainty and greater cue reliance, both when faced with the choice and when assessing confidence. Finally, confidence was related to action orientation, $F(2,110) = 10.72$, $p < .001$, $\eta_g^2 = .14$, showing that action-oriented participants were more confident ($M = 92.6\%$) than state-oriented ones ($M = 86.9\%$).

**DDM.** In the first stage of model selection, we compared 16 models with differentiating influence of cue type, stimulus on three model parameters: *z*, *v* and *Ter*. Responses were represented using a response coding scheme, where faces and houses represented the diffusion process's upper and lower bounds, respectively. This way, we can naturally interpret the drift rate parameter as sampling bias towards faces. We allowed starting point *z* to vary dependent on cue type. This would reflect strategic adjustments in how much evidence is needed; for

example, seeing an *80% house* cue might make participants shift their expectations so that they need less evidence to conclude the stimulus is a house and more to make a *face* choice. However, the starting point could not vary depending on stimulus type since that would require participants to have the ability to predict the stimulus before it was shown. Evidence accumulation *v* was allowed to vary both by cue and stimulus type, reflecting biased sampling induced by either cue or stimulus. Both of these effects were shown to play a role in previous studies [54, 72]. Finally, non-decision times could vary depending on cue since they have been previously linked to stimulus expectancy [73].

The model with separate drift-rate by cue type and stimulus and non-decision time by cue type produced the best fit (DIC score of -1819, 71-point difference compared to the second best). All parameters displayed excellent convergence (*R hat*<1.01). Simulating synthetic data from the model (posterior predictive check) validated its adequacy, showing that it could accurately reproduce quantile distributions of RT and accuracy data. Participants needed less evidence for the *face* response. The posterior distribution of *z* was significantly shifted towards face boundary $M = .62, P > .999$. Lack of influence of cue on starting point is surprising but consistent with some previous findings and theoretical speculations [72]. Faces were processed faster than houses $P (v_{\text{face}} > v_{\text{house}}) = .98$. Consistent with our behavioral results and previous findings [54], participants were influenced more strongly by the house cue. Cue-stimulus congruence significantly affected drift-rate for house stimuli ($P > .99$ for all comparisons), but not for faces. These findings are consistent with the accumulation process being biased not only by stimulus probability but also category reliability [54, 62, 74]. Stimulus category also influenced the speed of motor preparation and execution. Non-decision times were significantly faster for faces compared to houses, $M_{\text{face}} = 516$ ms, $M_{\text{house}} = 565$ ms, $P (Ter_{\text{face}} > Ter_{\text{house}}) = .98$, and trending towards being significantly faster than neutral, $M_{\text{neutral}} = 546$ ms, $P (Ter_{\text{face}} > Ter_{\text{neutral}}) = .91$.

In the second stage, we compared the winning stage 1 model (now referred to as *no-trait model*) with 7 additional models, varying the effects of action control on *z*, *v*, and *Ter*, and keeping the within-effect structure consistent with the *no-trait* model. Between-group differences in *z* and *v* would indicate cue-induced differences in information selection (see Table 2 hypothesis 1). Similar to the RDM task, there was very little variance DIC scores across the models (best DIC score = -1822, worst = -1819), which shows that adding additional between-group factor did not significantly improve model fit.

**Efficiency.**    A 3 x 2 mixed-design ANOVA (cue type x action orientation) on LISAS scores revealed a significant effect of cue, $F(1.11, 60.95) = 18.47$, $p < .001$, $\eta_g^2 = .09$. Participants were most efficient in the congruent and least in the incongruent trials. No evidence that efficiency varied across groups was found.

**Metacognitive accuracy.**    3 x 2 mixed-design ANOVA of meta-d' scores revealed a significant effect of cue type on metacognitive accuracy, $F(1.97, 108.53) = 14.98$, $p < .001$, $\eta_g^2 = .06$. Meta-d' was significantly lower when the cue was incongruent, compared to the other conditions. No other effects reached significance. Group differences in confidence magnitude and accuracies are presented in Fig 2.

## Discussion

Experiment 1 results paint a clear picture: confidence was the only measure consistently differentiating the groups, supporting the claim of differential post-decisional maintenance between groups (hypothesis 4). We found no support for hypotheses 1 to 3 postulating differences in the decision-making process. Crucially, even high difficulty accompanied by time pressure did not deteriorate decision-making performance of the state-oriented group, suggesting that, in

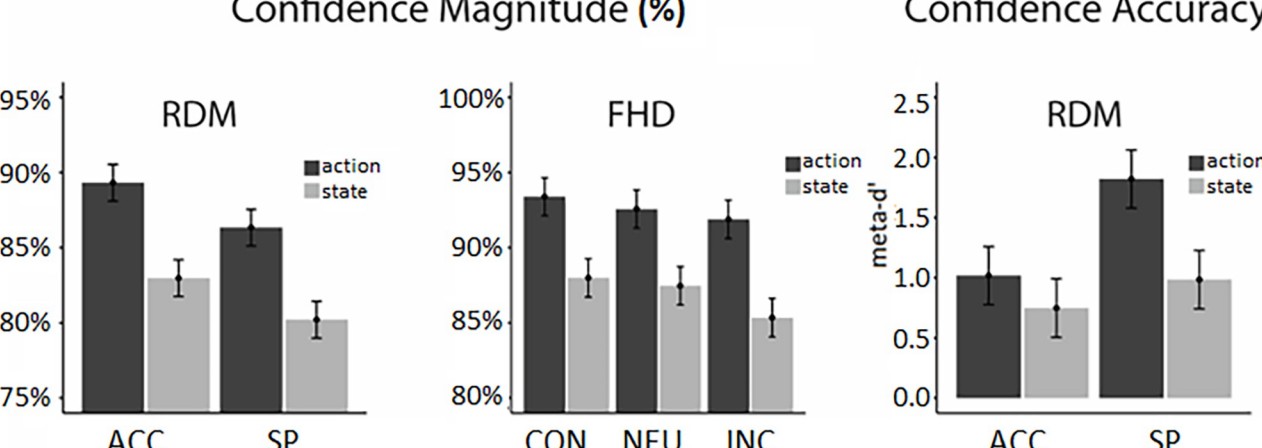

**Fig 2. Significant group differences in confidence magnitude and accuracy across the two tasks (RDM: Random dot motion; FHD: Face/house discrimination).** ACC and SP refer to the *accuracy emphasis* and *speed emphasis* conditions in the RDM task, respectively. CON, NEU and INC refer to the *congruent*, *neutral* and *incongruent* conditions in the FHD task, respectively. Error bars represent standard errors across participants.

the context of perceptual choices, volitional abilities shape behaviour by acting on the content of our choices only after we make them. This is further supported by additional analyses (LISAS) showing the groups did not meaningfully differ in terms of performance efficiency.

Among the findings, three points deserve special recognition. Firstly, the difference in metacognitive bias reflected in confidence magnitude was universal. The effect was independent of task content, difficulty, time pressure, or pre-choice bias. Moreover, magnitude effects were associated with the highest effect sizes among confidence predictors in both tasks ($\eta_g^2 =$ .17 and $\eta_g^2 = .13$ for RDM and FHD, respectively). This contrasts with the view that only highly demanding conditions should be affected by action control [9] and points to a more general mechanism of metacognitive evaluation.

Secondly, both groups expressed general overconfidence in the RDM task (mean confidence of 87.7% and 81.5% compared to mean accuracy of 67.4% and 65.2% for action and state-orientation, respectively) and an appropriate level of confidence in the FHD task (92.7% and 87.6% mean confidence compared to 90.4% and 88.2% mean accuracy). This finding is consistent with literature showing that overconfidence is more common in difficult tasks [75]. However, two caveats are necessary when assessing confidence levels in the FHD task. First, the lack of overconfidence might be partially due to the ceiling effect since the FHD task was performed at a very high level (S2 Table). Second, there was an ambiguity associated with the lowest confidence value: *50%*. In theory, it should reflect the lowest possible level of confidence, that is when participants make a random guess. In practice, however, it sometimes meant high confidence that the choice was incorrect. This could happen in trials where participants made a motor execution error or noticed an important detail during motor execution when they no longer could alter their choice. This is supported by the fact that the *50%* confidence level was chosen almost twice as often than *60%* (1644 vs. 985 times).

Thirdly, action-oriented individuals expressed a higher level of metacognitive accuracy in difficult task conditions. At first glance, this finding might appear intriguing. It seems counterintuitive that the action-oriented group was both more biased (overconfident) and more accurate at the same time. However, this is perfectly reasonable, as metacognitive accuracy relates to how well one can distinguish their correct responses from errors. As an illustration of this, someone who on average rates their correct decisions as 90% confidence and all their incorrect

decisions as 80% has a better insight than someone who rates both correct and incorrect at 60%, even though the second person might be closer to the actual average. Action-oriented participants were not just blindly overconfident in all their choices but also had greater insight into their correctness.

Together, Experiment 1 provides compelling evidence for the crucial role of post-decisional metacognitive processing in action control. These results align with action-driven theories of post-decisional processing, which postulate its central role in facilitating action [76]. Taking this further, we aimed at exploring this effect in more detail and addressing some of the limitations of the first study. Firstly, how universal is this effect? Current evidence is insufficient to claim that confidence is a universal mechanism distinguishing between action and state-oriented people. To explore this, we need to go beyond perceptual choices and compare different domains of decision making. Secondly, there are two possible mechanistic explanations for differences in post-decisional bias. First, *confirmation bias* assumes biased post-decisional sampling, where evidence confirming choice is associated with a greater gain [77]. This theoretical account predicts that the longer information is processed, the stronger the bias should be, as evidenced by recent findings [78]. Second, *biased read-out* assumes different mapping from decision-state to confidence assessment [79]. This account would predict similar differences in confidence, independent of how much time from the decision has passed. Finally, we might be interested, does task difficulty moderate the differences? Experiment 1 cannot directly answer this question, as the difficulty was only manipulated between tasks. Additionally the ceiling effect and an ambiguous confidence scale might have confounded some of the results. We aim to address these issues in Experiment 2.

## Experiment 2

In the second experiment, we aim to characterize the differences in metacognitive evaluation in more detail. First and foremost, whether the bias effect found in perceptual tasks generalizes to other domains. The issue of effect specificity is crucial because if the bias effect is only present in perceptual tasks, then generalizability to real-life scenarios might be limited. Previous studies on the domain-specificity of metacognitive judgements show mixed results. Correlations between bias and sensitivity across different perceptual tasks are well established [29, 80]. However, a meta-analysis of studies comparing metacognition between different domains (perceptual and memory) found only speculative proof of an association [81]. We hypothesize that differences in metacognitive processing between state and action-oriented people are a domain-general phenomenon and predict that the effect should hold for all choice types. To test this, we compare a perceptual task with a value-based one. We believe this makes for a stringent test of generality since the domains differ on two fundamental dimensions of the choice criterium: objectivity and whether it is externally or internally defined. Perceptual choices are associated with an objective, external criterion that defines the quality of responses. Value-based choices are driven by an internally defined criterion supported by subjective evidence. Previous studies comparing the domains suggest similar cognitive mechanisms associated with the decision process [82] and similarities between their neural signatures [83]. However, we are unaware of any studies comparing metacognitive processing between these two domains. To make the tasks comparable, we use a single set of stimuli while manipulating choice instructions. In the value-based choice task, participants selected one of two items based on their preference. In the perceptual task, participants decided which of the items was larger.

Our second goal is to describe the mechanism leading to differences in metacognition in more detail. Two possibilities are examined: biased post-decisional sampling and difference in

confidence read-out. Biased sampling hypothesis is an extension of the confirmation bias framework introduced by Brehm [84], which assumes that post-decisional processing privileges confirmatory evidence. Indeed, ample support was found for biased evidence weights and selection favouring the chosen option [77, 78, 85]. The need for cognitive dissonance reduction is often hypothesized as the main motivating factor of this bias [86]. If the degree of sampling bias differentiates the groups, we should expect to see a larger difference in confidence given more time to process the evidence after the choice is made. Alternatively, the difference might be associated with how choice uncertainty is read-out and mapped to explicit confidence judgements [79]. This mechanism is more general: if action control differentiates how people translate decision uncertainty into explicit metacognitive knowledge, one would expect to observe a stable difference independent of time spent on forming the judgement. A general, time- and domain-independent framework would provide a powerful tool in explaining differences in action enactment. In experiment 2., we test this by controlling for the interval between choice and confidence judgement.

Our third goal is to directly compare the effect of choice difficulty on metacognitive bias and accuracy. Experiment 1 revealed differences in accuracy in the difficult task and bias in both; however, the results are not directly comparable. Here, we manipulate decision difficulty as another factor. In line with a theoretical work proposed by Jostmann & Koole [9] and experimental results [87, 88], we predict the confidence gap to widen and judgement accuracy to decrease for state-oriented people as a function of choice difficulty. Instead of *free choice*, as in Experiment 1, we used an *interrogation* design, where participants responded only after hearing a signal prompting them to make a choice. This ensures that exposition times for each stimulus pair is equal and eliminates potential confound of decision RT affecting metacognitive accuracy (one might imagine that more cautious responding, and therefore longer processing time, could influence confidence judgements).

Based on these study goals, we derive the following hypotheses regarding the differences in confidence between groups (*confidence gap*) for Experiment 2.

*Hypothesis 1.* The confidence gap is domain-general. We expect to observe a between-group difference in confidence in both perceptual and value-based tasks.

*Hypothesis 2a.* The confidence gap is driven by biased sampling of evidence after the choice is made. This account predicts the gap to widen as a function of time given for consideration, after the choice is made.

*Hypothesis 2b.* The confidence gap is driven by biased read-out, manifested in biased sampling of evidence after the choice is made. This account predicts the gap to exist irrespective of consideration time.

*Hypothesis 3.* The confidence gap is affected by choice difficulty. We expect the gap to widen in the difficult choice condition.

Note that hypotheses 2a and 2b are not exclusive. If both accounts are true, we would expect to observe a significant difference in confidence in time point one (immediately after the choice), which then gets wider at time point two.

## Method

**Participants.**   An independent sample of participants was chosen from the same pool as in Experiment 1. Similarly to Experiment 1, we experienced ~50% rejection rate. The final participant pool for Experiment 2 consisted of 30 action-oriented participants (23 female and 7 male, $M_{age}$ = 21.86, $SD_{age}$ = 3.60; $M_{ACS\ score}$ = 18.1, $SD_{ACS\ score}$ = 1.82) and 26 state-oriented

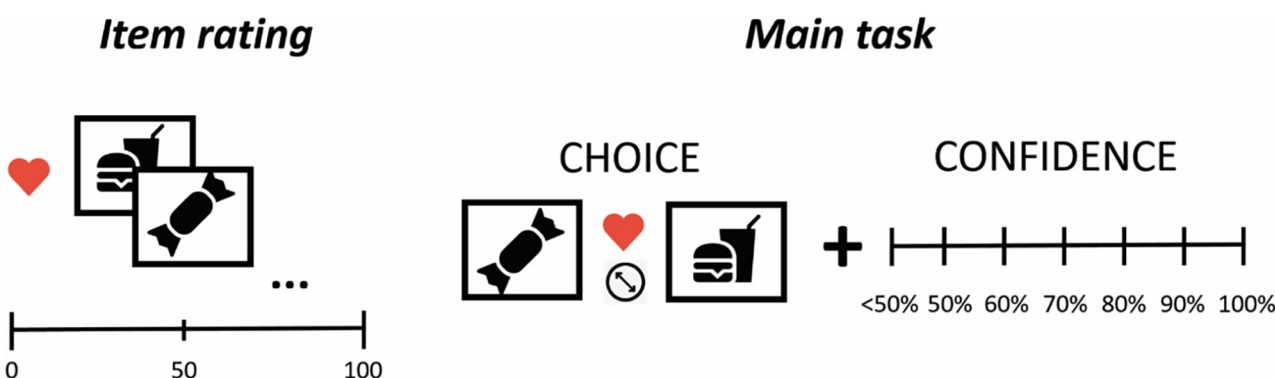

**Fig 3. Experiment 2 task design.** First, participants rated 72 food items based on their preference of winning the item. In the main task, each trial consisted of a forced choice between 2 items based on either participants' preference (value-based condition) or stimulus size (perceptual condition) followed by a confidence rating.

participants (25 female and 1 male, $M_{age}$ = 20.81, $SD_{age}$ = 1.67; $M_{ACS\ score}$ = 3.88, $SD_{ACS\ score}$ = 1.47). The standardized between-group difference in mean assessed against the distribution of ACS scores from the screening sample was equal to 2.64. The planned sample size for both groups was 30, however, 4 state-oriented participants failed to show up.

**Stimuli.** Stimuli consisted of 71 pictures of commercially available snacks displayed on a white background. Stimulus size was calculated by taking the proportion of non-white pixels in a 300 by 300 pixel image. Stimuli size distribution had a mean of 42.05% and a standard deviation of 16.46%, with the largest item taking 90.05% of the image surface area and the smallest 22.77%.

**Task.** The task consisted of an initial rating phase and a decision phase (Fig 3). In the first phase, participants rated their preference for each of the 71 items on a scale from 1 to 100. The main task consisted of repeated choices between two items and subsequent confidence ratings. The task consisted of 160 trials within a 2 x 2 x 2 (choice type x choice difficulty x choice-confidence interval) factorial design. In perceptual trials, participants decided which of the items is larger, while in the value-based condition they chose the item they would prefer to win. The instruction clarified that in the perceptual condition, *size* refers to the item surface area, and can be quantified as the percentage of non-white (item-related) pixels to white background pixels (so that an empty white image would have a size of 0, and an image where the item takes all the pixels would have a size of 100). A cue presented 500 ms before each choice indicated whether the next choice would be a perceptual or a value-based one. At the onset of the trial, two items were displayed on the left and right side of the screen. Participants could make a choice only after hearing a soft beeping sound played 2 seconds after the stimulus onset. Left and right responses were mapped to *z* and *m* keys, respectively. The confidence scale appeared either immediately or after 7 seconds of making a choice (long interval condition). The instruction elicited participants to spend the 7-second interval o thinking about one's confidence. The linear scale had 7 tick marks, under which the % confidence value was displayed. Values were mapped to keyboard keys from *0* to *6*, where *0* represented below 50% confidence (choice reversal) and *6* represented 100% confidence.

Choice pairs in the main task were constructed using an algorithm which 1) calculated the value difference (distance) between all available item pairs (n = 2485), 2) created *easy* and *difficult* choice groups by taking pairs where distance is greater than half of the maximum (easy) or smaller than 1/8 of the maximum, excluding pairs that are equal (difficult), 3) sampled 40 pairs

randomly out of each of the 2 groups. A similar procedure was repeated for perceptual choices; only here, the values were based on actual size.

Accuracy is operationalized as either choosing the larger (perceptual condition) or more preferred item (value-based condition). A limitation of accuracy analysis in the value-based condition is the assumption that the initial value ratings are noiseless, which is unlikely to be true. In reality, an item rated at 60% might be less preferred than one rated at 58%. As a result, accuracy might be a slightly inaccurate measure, especially in difficult trials. However, we do not expect this confound to meaningfully affect the inferences, since we have no reason to believe that the two groups differ in terms of the initial noisiness of item evaluation. To account for this possibility, we additionally compare the mean and the spread of initial ratings between the groups.

Trial sequence was randomized and counterbalanced across time interval factor and display order (left/right). Participants were rewarded with one, two or three items randomly drawn from the items they chose in the value-based condition. The number of rewards depended on the % of correct choices in the perceptual condition: scoring 90% or below—1 item; 90% to 95%—2 items; 95% or more—3 items. The reward scheme was constructed to motivate participants during both task types. Participants were also rewarded with course credits, independent of performance.

## Results

**Item value ratings.**    The ratings of the 71 items were distributed normally W = .989, $p$ = .84 ($M$ = 41.17, $SD$ = 9.38) with the highest rated item (a chocolate egg) scoring an average of 65.70, and the lowest (mint-flavoured caramel hard candy) an average of 16.57. Two-sided t-tests indicated no differences between group means $t(47.378)$ = 0.208, $p$ = .83 and standard deviations $t(51.624)$ = -0.815, $p$ = .42.

**Data pre-processing and analysis approach.**    We removed response times shorter than 200 ms and outliers over 3 standard deviations from the mean per subject and condition. Change of mind responses (indicated by <50% confidence; 1.4% of all trials) were also removed. We analyse each dependent variable using a mixed-design ANOVA model with within-person factors of task domain (perceptual or value-based), difficulty (easy or difficult), time interval (instant or delayed confidence judgement; used only in the analysis of confidence) and between-person factor of action orientation. In the value-based task, we define accuracy as choosing the option consistent with initial value judgements.

**RT and accuracy.**    Mixed-design ANOVA revealed a main significant effect of difficulty on RT, $F(1,54)$ = 123.92, $p < .001$, $\eta_g^2$ = .14. Responses in difficult trials were slower ($M$ = 642 ms) compared to easy choices ($M$ = 558 ms). There was also a close to significant effect of task type, $F(1,54)$ = 3.95, $p$ = .052, $\eta_g^2 < .01$, showing slightly faster responses during perceptual choices ($M$ = 594 ms vs. $M$ = 607 ms for value based). Analysis of accuracy indicated significant effects of difficulty, $F(1,54)$ = 508.79, $p < .001$, $\eta_g^2$ = .49, task type, $F(1,54)$ = 19.31, $p < .001$, $\eta_g^2$ = .12, as well as their interaction, $F(1,54)$ = 95.31, $p < .001$, $\eta_g^2$ = .15. Choices were more accurate in the easy ($M$ = 93.6%) and perceptual conditions ($M$ = 86.4%) compared to difficult ($M$ = 70.5%) and value-based ($M$ = 77.7%). The interaction effect indicates the difference between easy and difficult was more pronounced in value-based trials. Simple effect analysis indicated a significant difference in accuracy between easy and difficult trials in both tasks, $M_{perc}$ = 0.132, $t(108)$ = 9.123, $p < .001$; $M_{value}$ = 0.330, $t(108)$ = 22.883, $p < .001$, confirming that our difficulty manipulation was successful. High accuracies across all conditions suggest that participants were motivated to perform well. No effects of action orientation were observed. Lack of action orientation effects on choice supports the null findings of experiment one.

**Confidence.** Mixed-design ANOVA revealed significant effects of difficulty, $F(1,54) = 179.74$, $p < .001$, $\eta_g^2 = .31$, task, $F(1,54) = 12.89$, $p < .001$, $\eta_g^2 = .03$, and action orientation $F(1,54) = 11.29$, $p < .001$, $\eta_g^2 = .12$. Confidence was higher when choices were easy ($M = 93.3\%$) compared to difficult ($M = 82.6\%$) and when choices were value-based ($M = 89.2\%$) compared to perceptual ($M = 86.7\%$). An interaction between difficulty and time interval, $F(1,54) = 14.03$, $p < .001$, $\eta_g^2 < .01$, indicated a greater separation between easy and difficult choices after the interval. This means participant confidence after the interval decreased following difficult choices ($M = -1.08\%$, $t(104.2) = -2.478$, $p = .07$), while increasing after easy ones, $M = 1.00\%$, $t(104.2) = 2.289$, $p = .10$.

Action-oriented participants displayed higher confidence ($M = 90.8\%$) compared to state-oriented ones ($M = 85.1\%$). The difference between groups was more pronounced in the perceptual domain, $F(1,54) = 7.38$, $p < .01$, $\eta_g^2 = .01$, and when choices were difficult, $F(1,54) = 6.69$, $p = .01$, $\eta_g^2 = .02$. A three-way interaction between difficulty, time interval and action-orientation $F(1,54) = 7.18$, $p < .01$, $\eta_g^2 < .01$, revealed that the difficulty and interval interaction was driven by the state-oriented group. The gap in confidence between easy and difficult conditions grew in time only for state-oriented individuals. Simple effect analysis showed that when given more time for the confidence judgement, their confidence grew after easy choices, $M = 1.94\%$, $t(104.2) = 3.037$, $p < .02$, and diminished after difficult ones, $M = 1.63\%$, $t(104.2) = -2.548$, $p = .06$. In contrast, action-oriented participants displayed constant levels of confidence across the time interval (Fig 4).

**Metacognitive accuracy.** Metacognitive accuracy was higher after easy ($M = 2.856$) compared to difficult choices ($M = 0.594$), $F(1,54) = 44.85$, $p < .001$, $\eta_g^2 = .28$. This effect is expected, as metacognitive accuracy should by definition be affected by task difficulty [27]. Longer time interval also significantly increased judgement sensitivity ($M_{long} = 1.66$ vs. $M_{short} = 1.20$), $F(1,54) = 5.96$, $p = .02$, $\eta_g^2 = .04$, confirming the hypothesis that more time should be beneficial for accurate choice assessment. Neither task type nor action control significantly predicted metacognitive accuracy.

## Discussion

Results of the second experiment strengthen our initial findings and provide new insights into the confidence gap phenomenon. We were able to replicate the effect of post-decisional

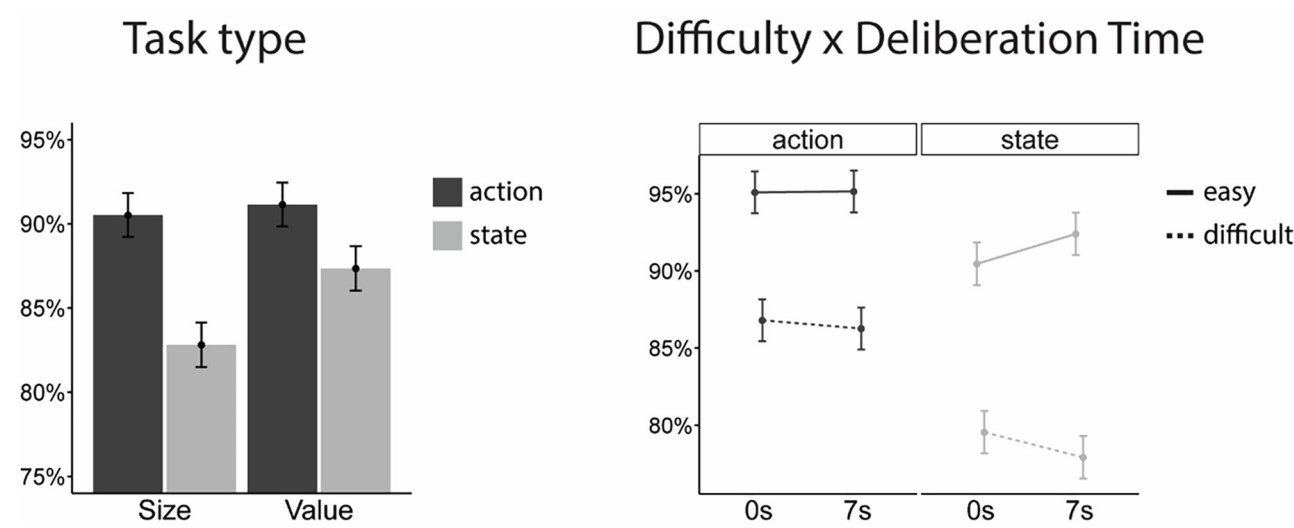

**Fig 4. Group differences in confidence magnitude.** Error bars represent standard errors across participants.

metacognitive bias acting as a distinguishing factor between action and state-oriented people. We also managed to generalize this result to value-based decisions (hypothesis 1). Choice difficulty proved to be an important factor, showing that the confidence gap widens when choices get tougher (hypothesis 3). Interesting insights also came from controlling the choice-judgement interval. As hypothesized, longer consideration led to more accurate judgements on a group level. Differences in confidence between groups were present at the initial time point and did not get larger as time passed, lending support to hypothesis 2b, but not 2a. The pattern that emerged was quite complex. We observed an increase in confidence for easy choices and a decrease for difficult ones among state-oriented individuals. In contrast, action-oriented individuals displayed a stable level of confidence across both time intervals. This finding has two possible interpretations. It could be that action-oriented people form their final confidence judgements immediately after committing to a choice and do not process any additional information given more time, while state-oriented people continue to ponder about it, effectively improving their judgement accuracy over time. This conclusion, however, stands in contrast with metacognitive sensitivity analysis, indicating that both groups express a noticeable improvement in judgement quality when given more time. The size of this improvement is similar for both groups, showing no indication that time works in favour of state-oriented people in particular. This leads us to believe that the difference in ratings between time intervals for state-oriented people is predominantly driven by task difficulty, not their actual accuracy. The difficulty is an external factor of choice at hand, while metacognitive accuracy reflects the quality of internal self-monitoring. By accounting for choice difficulty in their confidence ratings, state-oriented individuals did not gain better insight into their choices. Their judgements are (to a larger extent) a reflection of the external choice parameters. Interestingly, this line of reasoning is consistent with studies showing low action control being correlated with difficulties in distinguishing between internal cognitions and external cues [20, 88].

## General discussion

Action control constitutes a bridge between choices and actions. All too often our meticulous and carefully crafted plans—going to the gym, eating a healthy lunch, or writing a certain number of pages of a paper draft in a given day, fail due to a lapse in volitional control. Even though such lapses happen to almost everyone, some people are better at putting their goals into action than others, making it a stable individual difference.

Here, by rigorously testing choice and post-choice cognitive processing, we show that this trait is systematically related to post-decisional metacognitive processing. Action-oriented people display higher levels of choice confidence across task domains, choice difficulty, time pressure, and deliberation period. This confidence gap is dissociated from measures of performance that show no meaningful differences. Thus, higher confidence displayed by action-oriented people does not necessarily carry the negative burden of being less accurate in their assessments. In contrast, we found no meaningful differences in the decision-making process: when faced with fast-paced binary decisions, state-oriented people can respond as fast and accurately, adjust as flexibly to changing task demands, and incorporate additional information as parsimoniously as their action-oriented counterparts. The robustness of the *confidence gap* effect proves a pervasive link between metacognitive confidence bias and action control.

According to Kuhl's Personality System Interactions theory [3], differences in action control are driven by deficiencies in implicit affective processing. Specifically, the inability to inhibit negative affect (preoccupation dimension) and reduced ability to engage positive affect (hesitation dimension). Following this line of reasoning, many previous studies focused on manipulating factors influencing affect (for review see: Koole & Kuhl [89]). Our results expand

on the literature providing an across-domain mechanism shaping basic decision-making even in the absence of affective manipulation.

Another line of research focused specifically on the decision-commitment dynamic [90, 91]. Action-based model of cognitive dissonance proposed by Harmon-Jones, Harmon-Jones & Levy [41] postulates that action control influenced post-decisional processing, supporting the reduction of negative affect associated with dissonance via spreading of alternatives (boosting the value of a chosen option or devaluing the rejected one). Experiment 2 results did not support the hypothesis of diminished dissonance reduction in state-oriented individuals. Instead, they are more in line with a biased read-out account, where the same amount of evidence can be interpreted as more or less certain, depending on the individual's bias. Our data do suggest, however, that the content of post-decisional thoughts differs so that state-oriented persons weigh external cues more heavily when given time to assess confidence. This increased focus on external cues might be a compensatory mechanism for a diminished ability to access and integrate internal evidence, in line Kuhl's hypothesis of inhibited access to self-related memory [3].

We also found supportive evidence for the effects of high demand on metacognition—both bias and accuracy. Experiment 1 revealed differences in metacognitive sensitivity when the difficult task was combined with high time pressure. Interestingly, it was not the case that state-oriented participants' performance was diminished. Instead, high demand boosted the performance of action-oriented individuals, suggesting a better mobilization of cognitive resources under high demand [9]. In Experiment 2 we found that the confidence gap widens as a function of choice difficulty. These findings build on a large body of literature, showing that action control not only facilitates performance under pressure [37, 38, 92, 93], but also boosts the gap in subjective assessment.

## Beyond decision making

Confidence is often considered a stable self-belief construct [94] with a wide range of psychological and social benefits such as improving task motivation and persistence [95], maintaining high self-regard [75] and enhancing social status [96]. It is also a reliable predictor of academic achievement [97] and math accuracy across countries and cultures [98].

These findings overlap with the research on the effects of action control on well-being and achievement [6, 10, 11, 14]. Similarities involve not only outcomes but also postulated mechanisms. Individuals with low confidence tend to be less decisive [99], give up easier after failing (as described by the preoccupation dimension of action control), and have difficulty overcoming and navigating through daily tasks [100] (as described by hesitation dimension of action control). On top of that, both confidence magnitude judgements [29] and action control [3] are strongly associated with positive affect.

It is also important to contrast our findings concerning the confidence gap (and its known positive correlates) with results underlining detrimental effects of excessive confidence, such as those related to arrogance [101] or political radicalization [102]. Extreme levels of overconfidence might also lead to faulty assessments, unrealistic expectations and hazardous decisions [103], which suggests the possibility that the relationship between confidence and beneficial real-life outcomes is not monotonous. At the same time, it also might be worth noting that studies focusing on confidence calibration usually do not distinguish between bias and sensitivity [104–106], making the measure susceptible to being influenced by both factors. Detrimental overconfidence requires both—a high bias, *i.e.* a tendency to display high levels of confidence in general, as well as poor sensitivity, *i.e.* the inability to change one's mind in light of contradicting evidence.

In addition, in our study, both groups displayed characteristic patterns of overconfidence when the task was easy and underconfidence when it was difficult [76] (*Supplementary Materials*) highlighting that the difference was not qualitative, but quantitative in nature.

Taken together, these findings give credence to the hypothesis that the confidence gap might not only be associated with differences in simple decision making, but also a crucial factor underlying differences in action control. This perspective could shift how we think of action control as a construct. Classical accounts most often frame it as a failure of the executive system. While it is true that commitment in the face of uncertainty is inhibited in state-oriented people [107], our data suggest that it is more due to relative overestimation of uncertainty and not necessarily the lack of commitment skills. Low subjective confidence can inhibit action commitment, therefore freezing action plan preparation [107].

## Cognitive underpinnings of confidence

Our results are consistent with the view that confidence is a *common currency* transcending task domains and responsible for comparing accuracy across different decision types [108]. A unique mechanism dedicated to supervising the quality of our choice or action is necessary for error detection and changes of mind [109–111]. In contrast to highly specialized decision-making faculties, it requires a common encoding core with dedicated brain networks [112].

Confidence tracks evidence present both before and after the decision. Boldt, Schiffer, Waszak and Yeung [113] found that confidence predictions before choice were highly indicative of post-choice assessments. The latent confidence variable shares variance with the decision variable (due to shared task evidence) but also has its unique variance (a different encoding mechanism and additional evidence it can sample before and after the choice is made). Such specification enables confidence to flexibly encode the level of uncertainty given decision evidence and a range of relevant meta cognitions such as expected task proficiency, memories of previous experiences, or general self-efficacy. Recently few candidate models have been proposed to explain how confidence arises, some of which focus on differences in sampling [114–117], differences in uncertainty mapping [79, 117], or both [24]. Our work lays the groundwork for testing and relating these mechanistic models to personality differences. Such work would benefit cognitive-focused research—by applying the models to relevant human behaviour and comparing them in terms of predictions and explanatory power. It could also inspire personality-focused projects—by getting a better understanding of the underlying differences driving personality differences.

## Limitations and future directions

Our study leaves an open door for further exploration of some of the findings. Firstly, our results provide non-conclusive evidence for differences in metacognitive accuracy, which were not replicated in Experiment 2. A likely reason for this is not including time pressure manipulation, which resulted in the demand not being high enough to induce differences in observed behaviour. Nonetheless, we believe this effect would benefit from further investigation.

Secondly, our tasks comprise only a slice of the spectrum of decision-making paradigms, let alone real-life decisions. It is crucial to stress that we do not claim nor believe that action control does not influence decisions, especially when choices are complex or rely on previous actions. While our study focused explicitly on task design that dissociated decision-related processes from confidence, future work should also consider cases where metacognitive judgements can influence subsequent choices. Realising meaningful life goals requires complex multi-stage processing, where previous choices influence subsequent ones. In scenarios where choices constitute a dependent sequence, metacognitive assessment of intermediate steps

should influence future decisions. A similar framework would also test the mediatory role of confidence between action control and commitment.

Thirdly, a concern might be raised that method was not sensitive to find reliable differences in primary task performance. Indeed, we focus on finding at least medium-sized effects. Our tasks have 80% power to detect effects of Cohen's d = 0.64 or larger. In practice, our model-derived measures can likely detect reliably slightly smaller effect sizes as well, since Hierarchical Bayesian approach is associated with increased efficiency in detecting group differences [118], as are extreme group designs [119]. To account for this, we performed an additional simulation-based analysis (see: Supplementary Materials) and found we had over 80% power to detect a between-group difference in any given task or condition if the variable of interest is correlated with the ACS score variable at r >= 0.3. Given that our measures of interest were tested across different conditions and experiments, the power to detect such difference in at least one instance is considerably greater. We believe that our null findings in primary task performance across three tasks, two samples of participants and a multitude of performance measures give reasonable support that if such effects exist, they are small and relatively insignificant compared to the differences in confidence.

Fourthly, given the limited scope of the study (limited sample, set of experimental tasks and measures used), we cannot fully rule-out some alternative explanations. *E.g.* performance in a controlled laboratory environment might not be ecologically valid and does not always correspond to real-life situations [120]. Another factor is that confidence ratings are based on self-report scale, making it susceptible to additional confounds. In this light, *state-oriented* individuals might only express lower confidence while not truly experiencing it. Such *report bias* [121] could be driven by factors including modesty or excessive self-monitoring [122]. Finally, a more mechanistic account of metacognitive bias on cognitive and neural levels is needed to understand how the confidence gap arises. A recent prominent framework proposed by Fleming and Daw [24] suggests that confidence is assessed by conditioning the confidence variable on choice. Metacognitive accuracy is determined by the quality of the confidence variable and the tightness of coupling between the decision and confidence variables, while bias—by the *beliefs* about these parameters. In this view, the beliefs constitute hyperparameters determining how evidence translates into confidence magnitude. This interpretation gives way to move from inferring about a psychologically salient but mechanistically murky concept of confidence magnitude to experimentally tailored manipulations targeting specific parameters of the model. Together, these considerations outline the limits of the generalizability of our findings.

## Conclusion

This paper compared decision-making mechanisms between action and state-oriented people using paradigms capable of dissociating between decision-making mechanisms and confidence processing. Our results suggest that the differences between groups arise not due to how evidence is obtained or strategic adjustments but due to how it is interpreted by a higher-level process relating evidence to subjective confidence.

Since confidence has been shown to influence future choices [123], this interpretation can be extended to more complex problems where strictly decision-related processes such as accumulating, weighing, and comparing evidence in favour of different actions are intertwined with metacognitive evaluations of the quality of the evidence. Most human goals involve a sequence of choices and actions. For example, planning to open a restaurant requires obtaining funding, choosing and renting a location, getting permits and licences, hiring staff, preparing an advertising campaign and more. At each point in this multi-stage process, our confidence can influence how likely we will progress. Since completing all the intermediate

steps is necessary to succeed, low confidence bias has many opportunities to seed doubt and lead to giving up. It is easy to see how this can generalize to more trivial matters such as preparing for an exam, exercising, or learning a new skill, where low confidence in meaningful progress can discourage a regular practice. Further research using different tasks and measures is necessary to confirm these speculations and extrapolate the generalizability of our findings to domains outside of perception and value-based decision tasks. All else being equal, this confidence gap might be the reason why some people succeed where others cannot.

## Supporting information

**S1 Fig. Confidence calibration curves in the RDM task.** ACC: accuracy emphasis condition. SP: speed emphasis condition.
(TIF)

**S2 Fig. Confidence calibration curves in the FHD task.** CON: congruent cue condition; NEU: neutral cue condition; INC: incongruent cue condition.
(TIF)

**S3 Fig. Confidence calibration curves in Experiment 2 for value-based choices.** easy: easy condition; hard: difficult condition; 0.05: short choice-confidence interval condition (0.05 sec); 7: long choice-confidence interval condition (7 sec).
(TIF)

**S4 Fig. Confidence calibration curves in Experiment 2 for perceptual choices.** easy: easy condition; hard: difficult condition; 0.05: short choice-confidence interval condition (0.05 sec); 7: long choice-confidence interval condition (7 sec).
(TIF)

**S5 Fig. Power of detecting a significant result (at a $p < 0.05$ level) given true correlation level between ACS score and a variable of interest in our extreme-group design.** Dashed horizontal red line indicates 80% power level.
(TIF)

**S1 Table. Marginal means and standard deviations for 3 observable dependent measures across tasks in Experiment 1: Accuracy, reaction times (RT) and confidence, averaged across subjects, conditions and action orientation (Action vs State).**
(DOCX)

**S2 Table. Marginal means and standard deviations for 3 observable dependent measures across tasks in Experiment 2: Accuracy, reaction times (RT) and confidence, averaged across subjects, conditions and action orientation (Action vs State).**
(DOCX)

**S3 Table. Correlations between 4 variables of interest (RT, accuracy, confidence and meta-d' for the accuracy emphasis condition in the RDM task).**
(DOCX)

**S4 Table. Correlations between 4 variables of interest (RT, accuracy, confidence and meta-d' for the speed emphasis condition in the RDM task).**
(DOCX)

**S5 Table. Correlations between 4 variables of interest (RT, accuracy, confidence and meta-d' for the congruent condition in the FHD task).**
(DOCX)

**S6 Table. Correlations between 4 variables of interest (RT, accuracy, confidence and meta-d' for the neutral condition in the FHD task).**
(DOCX)

**S7 Table. Correlations between 4 variables of interest (RT, accuracy, confidence and meta-d' for the incongruent condition in the FHD task).**
(DOCX)

**S8 Table. Correlations between 4 variables of interest (RT, accuracy, confidence and meta-d' for the value-based condition in Experiment 2).**
(DOCX)

**S9 Table. Correlations between 4 variables of interest (RT, accuracy, confidence and meta-d' for the perceptual condition in Experiment 2).**
(DOCX)

**S10 Table. Correlations between 4 variables of interest (RT, accuracy, confidence and meta-d' for the easy condition in Experiment 2).**
(DOCX)

**S11 Table. Correlations between 4 variables of interest (RT, accuracy, confidence and meta-d' for the difficult condition in Experiment 2).**
(DOCX)

**S12 Table. Correlations between 4 variables of interest (RT, accuracy, confidence and meta-d' for the short-interval condition in Experiment 2).**
(DOCX)

**S13 Table. Correlations between 4 variables of interest (RT, accuracy, confidence and meta-d' for the long-interval (7 seconds) condition in Experiment 2).**
(DOCX)

**S14 Table. An additional power analysis based on a simulation approach based on the following procedure.** 1) We simulate a variable of interest, correlated with the ACS scale at a given level 2. We randomly selected 60 participants from our pool to the 2 extreme groups, with a minimum combined score of 25 for the action-oriented group, and a maximum combined score of 6 for the state-oriented group (similar to the actual groups used in the experiments). 3. We perform a 2-sample t-test, to see if the difference between groups is detectable at a $p < 0.05$ level. We then run 10000 simulations for each level of correlation (from 0.05 to 0.5, in 0.05 increments). Power can be calculated by subtracting the fraction of false positive findings from all significant results, at each level.
(DOCX)

## Acknowledgments

We thank Martjin Mulder (University of Amsterdam) and Kyle Dunovan (Carnegie Mellon University) for sharing code and advice for task procedures in experiment 1, and Monika Dąbkowska for her help in data collection.

## Author Contributions

**Conceptualization:** Wojciech Zajkowski, Magdalena Marszał-Wiśniewska.

**Data curation:** Wojciech Zajkowski.

**Formal analysis:** Wojciech Zajkowski, Maksymilian Bielecki.

**Funding acquisition:** Wojciech Zajkowski.

**Investigation:** Wojciech Zajkowski.

**Methodology:** Wojciech Zajkowski, Maksymilian Bielecki.

**Project administration:** Wojciech Zajkowski.

**Supervision:** Magdalena Marszał-Wiśniewska.

**Visualization:** Wojciech Zajkowski.

**Writing – original draft:** Wojciech Zajkowski, Maksymilian Bielecki, Magdalena Marszał-Wiśniewska.

**Writing – review & editing:** Wojciech Zajkowski, Maksymilian Bielecki, Magdalena Marszał-Wiśniewska.

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
