## [Decision Letter · Decision Letter 0]

6 May 2021

PONE-D-21-08854

Are you confident enough to act? Individual differences in Action Control are associated with post-decisional metacognitive bias

PLOS ONE

Dear Dr. Zajkowski,

Thank you for submitting your manuscript to PLOS ONE. After careful consideration, we feel that it has merit but does not fully meet PLOS ONE’s publication criteria as it currently stands. Therefore, we invite you to submit a revised version of the manuscript that addresses the points raised by the reviewers.

We look forward to receiving your revised manuscript.

Kind regards,

Eugene V Aidman, PhD (Psychology)

Academic Editor

PLOS ONE

Journal Requirements:

Please include captions for your Supporting Information files at the end of your manuscript, and update any in-text citations to match accordingly. Please see our Supporting Information guidelines for more information: http://journals.plos.org/plosone/s/supporting-information.

Please ensure that you refer to Figure 3 in your text as, if accepted, production will need this reference to link the reader to the figure.

Reviewers' comments:

Reviewer's Responses to Questions

**Comments to the Author**

1. Is the manuscript technically sound, and do the data support the conclusions?

Reviewer #1: Partly

Reviewer #2: Yes

2. Has the statistical analysis been performed appropriately and rigorously? 

Reviewer #1: Yes

Reviewer #2: Yes

3. Have the authors made all data underlying the findings in their manuscript fully available?

Reviewer #1: Yes

Reviewer #2: Yes

4. Is the manuscript presented in an intelligible fashion and written in standard English?

Reviewer #1: Yes

Reviewer #2: Yes

5. Review Comments to the Author

Reviewer #1: This area of research is of much interest, and the paper is aimed at gaining a deeper understanding of how individual differences in action control relate to metacognitive bias, here over- and under-confidence, based on confidence judgements embedded in the cognitive act. While this manuscript has merit, in its current state it suffers from a number of problems which I hope the authors will be able to address. Overall, I support this line of research, and think that it adds important novel information to and clarifies the existing body of knowledge.

One of my main concerns is the author’s lack of knowledge of research which already goes on in this area. Below, I list just a few papers, but there is a larger body of research available in the literature. This compromised the authors’ use of terminology, which is inconsistent with the main body of literature and some of their claims. My other concerns are with the method. Some of the issues can be worked around by declaring them in the limitation sections, but some issues are more serious. Please see my comments below.

Introduction and literature review:

1. The authors want to examine the “cognitive mechanisms responsible for generating choices and post-decisional processing” p. 4. They used confidence judgements embedded into performance on various cognitive tasks, which they labelled as “post-decisional confidence”. I must admit that I struggled with the use of this label as much of the metacognitive literature I’m familiar with, use different terms. Importantly, this, more accepted terminology is also described in the Fleming SM, Lau HC. How to measure metacognition. Front Hum Neurosci. 2014 Jul 15;8:443 cited by the authors. It includes ‘confidence judgements’ instead of ‘post-decisional confidence’, ‘discrimination’ instead of long-handed explanations provided by the authors about a need of having higher confidence for correct answers and lower confidence for incorrect answers, sometimes referred to as confidence magnitude and accuracy (sensitivity), sometimes referred to as “accuracy” [p. 24, line 580] metacognitive accuracy [see p. 24, line 581]. I strongly encourage authors to use terminology consistent with the previous research done via decision-making and metacognition/learning paradigms. I also encourage the authors to present relevant calibration curves for each test.

2. I doubt that Dibbelt and Kuhl would use confidence judgements to capture “post-decisional maintenance processes” described in their theory. In general, I cannot help but question the authors choice to operationalise Dibbelt and Kuhl’s model using the metrics selected. Currently, the provided justifications are very relaxed (see p. 7). Please provide more detail justifications.

3. I also encourage the authors to create a table stating definitions and relevant formulae for ALL metrics used in this study. The later was a point of much confusion to me as Table 1 was not sufficient enough. E.g., I was not sure whether the authors used absolute difference between confidence and accuracy (referred to on p. 4) or a raw difference, which would also capture under-confidence.

Some other questions.

The authors state “The association between bias and accuracy is complex. Both extremely high and low bias would lead to poor accuracy (very low discriminatory power between correct and incorrect choices). However, apart from extreme cases, the dimensions become more independent, as documented by dissociating effects found in the aforementioned studies.” P. 5.

I find this statement to be misleading. The dimensions do not become more independent as there is a straightforward mathematical dependency between bias and accuracy.

Papers based on the individual differences approach of direct relevance to this ms (more are available):

Jackson, S. A., Kleitman, S., Stankov, L. & Howie, P. (2017). Individual differences in decision making depend on cognitive abilities, monitoring and control. Journal of Behavioral Decision Making, 30(2), 209-223.

THIS PAPER TALKS ABOUT THE CRUCIAL LINK BETWEEN CONFIDENCE JUDGEMENTS AND ACTION CONTROL.

Kleitman, S., & Stankov, L. (2007). Self-confidence and metacognitive processes. Learning and Individual Differences, 17(2), 161–73.

Stankov, L & Crawford, J. D. (1997). Self-confidence and performance on tests of cognitive abilities. Intelligence, 25(2), 93-109.

BOTH PAPERS ADDRESS MEASUREMENT ISSUES OF CONFIDENCE AND ITS CALIBRATION.

Method/Results:

Why these sample sizes were selected in each study and for each experimental condition (n=30)? And why more participants were not recruited when some did not show up for the second experimental sessions?

In results, the authors should report descriptive statistics for ALL of the metrics used, including the estimates of their internal consistency whenever appropriate.

Also, was the accuracy of performance analysed in a separate analysis? Were these results presented?

An optional suggestion: I would also like to see correlations between different metrics, possibly in appendices/sup materials. I’m painfully aware of small sample sizes used in both studies; but still, these estimates might provide further clarifications of what is going on.

Also, was there an effect of gender?

Discussion in Experiment 1:

The authors state “Confirmation bias assumes biased post-decisional sampling, where evidence confirming choice is associated with a greater gain [78]. This account predicts that the longer information is processed, the stronger the bias should be” (p. 25, lines 595-97). I do not agree with this stipulation, as longer information processing can also indicate that the individual is considering other options, activating system 2, thus resulting in smaller confirmation bias.

The authors also state that “Additionally, some of the results might have been confounded by the ceiling effect and an ambiguous confidence scale” (p. 25, lines 601-2). These two points are unclear. Providing descriptive statistics can help to clarify the existence of ceiling effect. I’m not clear, however, why the authors claim that that the confidence scale used in Experiment 1 was ambiguous? And how was it different in Experiment 2?

Experiment 2

The authors state:

“Perceptual choices are associated with an objective, external criterion that defines the quality of the choice. Value-based choices are driven by an internally defined criterion supported by subjective evidence. Previous studies comparing the domains suggest similar cognitive mechanisms associated with the decision process [82] and some similarities between their neural signatures [83]. However, we are unaware of any studies comparing metacognitive processing between the two”. P. 26, lines 618-623.

This statement is incorrect as there is extensive previous research available. E.g., (and there are more):

Kleitman, S., & Stankov, L. (2001). Ecological and person-driven aspects of metacognitive processes in test-taking. Applied Cognitive Psychology, 15, 321–41. doi:10.1002/acp.705

Stankov, L. (1998). Calibration curves, scatterplots and the distinction between general knowledge and perceptual tasks. Learning and Individual Differences, 10(1), 29-50.

Method/Results:

Please justify sample sizes and provide clarifications specified for Experiment 1 (e.g., table with descriptive statistics etc., calibration curves). Were there gender differences?

Discussion/general discussion

“Our results expand on the literature providing a universal mechanism shaping basic decision-making even in the absence of affective manipulation.” (p. 34). I find the use of the word “universal” to be a bit inflated, especially given the limitations described above. Also, what seems to be missing for me is a discussion of how despite using completely different method and analytics, some of these results extend and replicate the results obtained via using the individual differences paradigm.

Below is some additional feedback from one of my senior PhD students whose research focuses on confidence, its calibration and control aspects. I endorse their comments below and think they should help the authors with their revisions.

Reviewer #1a:  "This is an interesting paper which compares action control to cognitive and metacognitive constructs within decision-making. The paper investigates an important research area, combining individual differences in cognitive and metacognitive processes with decision-making processes.

Some comments:

1. There seems to be considerable differences between the theories surrounding High and Low Action Control individuals and the results of the current experiment. It appears that much of the Introduction is on how action-oriented individuals have beneficial attributes, particularly those surrounding flexible decision-making in more challenging situations whilst state-oriented individuals cannot adjust plans and will fail in their decision-making. From the results, it appears that the only difference between the two groups is that one is more confident than the other. The authors mention that without a time pressure manipulation, it was hard to induce differences. However, I don’t see how this is a limitation considering participants had generally responded to items within 1 second.

2. The action control variable appears to be similar in definition to adaptability and resilience, specifically within the decision-making domain. It would be good to mention this to some extent.

3. Although I understand the benefits of confidence, particularly those stated in the conclusion, it is important to point out the differences in metacognitive accuracy, or lack thereof. The current study indicates that there is either a greater or similar metacognitive accuracy comparing action-oriented and state-oriented groups. In this sense, it’s important to note that action-oriented individuals appear to have greater confidence without the negative aspect of overconfidence.

Minor Comments:

- In the introduction, the authors mention that Action Control is related to differences in cognitive control and ability citing moderate positive correlations with consciousness and self-monitoring. Although I understand the self-monitoring relationship with cognitive control, I don’t see any mention of the relationships there with cognitive ability (i.e., intelligence).

- In the 2nd experiment, are participants allowed to rate preferences similarly across the items (e.g. a preference for 80 for both)? If this is the case, how would accuracy be computed if the two are paired with each other?

- Please label the y axes of the Experiment 2 figures."

Reviewer #2: The paper presents a well-designed, novel, and interesting study with results that could potentially make an important contribution to the literature. The drawbacks of the study include small sample size which may have limited the possibility to detect between-subject effects and their interactions, as well as complexity of the paper, which makes it difficult to understand and appreciate the findings. As a general recommendation to improve the paper readability, I would suggest to make more connection in the discussion to the original hypotheses, and more comparison and connection of the findings of the two experiments. I would like to point out some additional minor points in the text where additional detail could benefit the presentation.

Line 104: “The association between bias and accuracy is complex. Both extremely high and low bias would lead to poor accuracy” : perhaps, saying “strong positive” and “strong negative” instead of “high” and “low” would be clearer.

Line 110: Adding a few words to clarify the notion of “alternative spreading” for readers familiar with the concept, but not the term, would be good (e.g., “alternative spreading (opposite changes in the attractiveness of the preferred and the competing alternatives)”)

Line 258-260: It might be good to quantify the magnitude of the difference between the contrast groups (as Cohen’s d) here and in Experiment 2, to make it clear how huge the effect size is, thanks to the large initial sample.

Line 280: It would be better to report the correlations between the ACS subscales in the present sample (N = 724), either here or below (Line 396).

Line 310: From the description of Procedure it is not clear whether the study was carried out in a lab setting or online.

Line 355: “Conventionally, differences larger than 10 in DIC scores are considered significant [61]” : this appears to simplify things a bit too much, given that Spiegelhalter et al. (2002, p. 613) write: “Burnham and Anderson (1998) suggested models receiving AIC within 1–2 of the ‘best’ deserve consideration, and 3–7 have considerably less support: these rules of thumb appear to work reasonably well for DIC.” Based on this, the authors’ approach to interpreting DIC appears to be an oversimplification with ramifications for model choice: “All the models displayed very similar DIC scores (between 8441 and 8448), indicating that adding action control as a between-person factor did not significantly improve model fit.” (Line 445) : it looks like the models with a DIC difference of 7 can hardly be deemed equivalent (although several models with DIC 8441-8443 can be?)? Because this rule of thumb might have substantive consequences, a more careful treatment of DIC seems necessary. I would suggest not to interpret the differences in DIC between models as “significant”, but rather in terms of relatively better/worse model fit.

Line 423: “Condition-dependent” probably needs a dash.

Lines 420-430: A description of DDM parameters might be better placed to the Methods section.

Lines 542-543: The legend for Figure 2 needs to explain all the abbreviations (ACC, SP, CON, NEU, INC).

Line 546: It would be good if the results and discussion of Experiment 1 made reference to the numbered hypotheses presented at the beginning.

Line 588: “These results align with action-driven theories of post-decisional processing.” An explanatory sentence describing the predictions of these theories would be good here.

Line 654: It would be great to present the study goals in a more structured way of numbered hypotheses, as in the description of the first experiment.

Line 672: How exactly was the task formulated for participants? The text mentions “size” and “surface area”, which might not be completely interchangeable.

Line 685-686: The preference ratings are based on each object being rated by a participant on a 100-point scale and just once. The following procedure using these individual ratings to generate easy/difficult pairs treats preference ratings as perfectly measured. This approach is totally fine for size, which is an objective property, but for value one can expect some measurement error (which might be fairly large, given the single-item nature of the measure). For “difficult” pairs, the distance between two items being compared might actually be smaller than the uncertainty associated with the error of measurement (which, unfortunately, is unknown). As a result, an item rated 50% might really be less preferred than one rated 48%. The untenable assumption of perfect measurement makes it hardly possible to interpret accuracy ratings for values (at least for difficult choices), which needs to be reflected in the presentation of the results and discussion.

(As an additional possibility, it appears that individuals did not have a time limit and could change their answers, which may have introduced some metacognitive bias, especially for state-oriented individuals. Could it be that the size and the distribution of measurement error in deriving initial value ratings was different in action and state-oriented individuals?)

6. PLOS authors have the option to publish the peer review history of their article (what does this mean?). If published, this will include your full peer review and any attached files.

Reviewer #1: No

Reviewer #2: No

---

## [Author Response · Author response to Decision Letter 0]

20 Jun 2021

Dear Professor Aidman,

Thank you for your and the reviewers’ attention and comments on our manuscript. We are pleased that the reviewers have found our work of general interest and importance. The specific comments have been helpful and constructive in the revision of the manuscript. We have addressed each of the issues raised by the referees and revised the manuscript accordingly. Below we summarize our response to the comments, as well as the changes implemented in the revision. Reference section contains additional literature cited in this response. The line numbering indicated in the responses below relates to the revised manuscript with tracked changes highlighted.

Yours sincerely,

Wojciech Zajkowski, Maksymilian Bielecki, Magdalena Marszał-Wiśniewska

Reviewer 1.

Introduction and literature review:

1. The authors want to examine the “cognitive mechanisms responsible for generating choices and post-decisional processing” p. 4. They used confidence judgements embedded into performance on various cognitive tasks, which they labelled as “post-decisional confidence”. I must admit that I struggled with the use of this label as much of the metacognitive literature I’m familiar with, use different terms. Importantly, this, more accepted terminology is also described in the Fleming SM, Lau HC. How to measure metacognition. Front Hum Neurosci. 2014 Jul 15;8:443 cited by the authors. It includes ‘confidence judgements’ instead of ‘post-decisional confidence’, ‘discrimination’ instead of long-handed explanations provided by the authors about a need of having higher confidence for correct answers and lower confidence for incorrect answers, sometimes referred to as confidence magnitude and accuracy (sensitivity), sometimes referred to as “accuracy” [p. 24, line 580] metacognitive accuracy [see p. 24, line 581].

 I strongly encourage authors to use terminology consistent with the previous research done via decision-making and metacognition/learning paradigms. I also encourage the authors to present relevant calibration curves for each test.

Regarding the term post-decisional confidence

The term post-decisional confidence was used in order to distinguish it from prospective judgements of confidence (Fleming et al. ,2016), such as confidence predictions (Boldt et al., 2019) or feeling of knowing (Elman et al., 2012). Cited literature does refer directly to post-decisional biases in confidence (Navajas, Bahrami & Letham, 2016), and we believe our use of the term is justified, especially since it provides a link with post-decisional maintenance process, described by Dibbelt & Kuhl (for an in-depth explanation also see the reply to the second comment). 

In order to clarify this, we provide an explicit explanation:

lines 92-94:

In reference to confidence judgements, we use the term post-decisional as opposed to more frequently encountered term retrospective [26] to accentuate the role of choice in shaping the metacognitive process.

Regarding the ‘accuracy’ issue:

We always refer to the quantity measured by the meta-d’ parameter as ‘metacognitive accuracy’ or ‘metacognitive sensitivity’. The term ‘accuracy’, without ‘metacognitive’ preface mentioned in the comment refers to actual task accuracy (whether the response was correct or not), not metacognitive accuracy. In order to keep the terminology consistent, we switched all instances of ‘metacognitive sensitivity’ to ‘metacognitive accuracy’, as well as added a clarification.

lines 101-103:

 Metacognitive accuracy is the sensitivity of confidence judgements (sometimes referred to also as metacognitive sensitivity [28])

Regarding the term discrimination as a substitute for a long-handed explanation:

While the term ‘discrimination’ is a concise way of describing metacognitive sensitivity, we believe that a more long-winded explanation is a helpful inclusion for readers that do not have much experience in using signal detection theory-based analyses.

To make it clear that we are in fact describing discriminability, we modify the following

lines 103-107:

More accurate judgements mean a better insight into the decision process (higher discriminability). A highly accurate agent would have relatively low confidence in their incorrect choices and relatively high in correct ones. Accuracy was shown to decline with aging [30] and to be consistently subpar among people holding radical beliefs [31]. 

Regarding the calibration curves:

As requested, we add confidence calibration curves in the Supplementary Materials (section Confidence calibration curves).

2. I doubt that Dibbelt and Kuhl would use confidence judgements to capture “post-decisional maintenance processes” described in their theory. In general, I cannot help but question the authors choice to operationalise Dibbelt and Kuhl’s model using the metrics selected. Currently, the provided justifications are very relaxed (see p. 7). Please provide more detail justifications.

Post-decisional maintenance refers to decision stability in time. Consistent with this understanding, retrospective measurement of confidence is a proxy of the post-decisional maintenance process, reflecting the certainty associated with the decision and influencing the likelihood of implementation. In fact, Dibbelt & Kuhl (1994) use confidence ratings measuring post-choice maintenance (p.189), when asking participants to rate their post-choice satisfaction with a chosen job offer.

To make this connection more clear, we added the following

lines 159-162:

Finally, decision maintenance (4) can be related to post-decisional confidence judgements, which reflect the certainty associated with the decision and influence the likelihood of implementation, as predicted by Kuhl’s model [36]. 

3. I also encourage the authors to create a table stating definitions and relevant formulae for ALL metrics used in this study. The later was a point of much confusion to me as Table 1 was not sufficient enough. E.g., I was not sure whether the authors used absolute difference between confidence and accuracy (referred to on p. 4???) or a raw difference, which would also capture under-confidence.

Table describing the model-derived parameters of interest in more detail have been added (Table 1; lines 179-182)

Both accuracy and confidence were analysed separately. The only difference variable we used was Δa, referring to the difference between the threshold parameter ‘a’ between the two conditions of the RDM task. To make this clear, we adjusted the parameter description:

lines 256-257:

High flexibility would be associated with a large difference in threshold (Δa) between accuracy and speed conditions (aaccuracy – aspeed).

Some other questions.

The authors state “The association between bias and accuracy is complex. Both extremely high and low bias would lead to poor accuracy (very low discriminatory power between correct and incorrect choices). However, apart from extreme cases, the dimensions become more independent, as documented by dissociating effects found in the aforementioned studies.” P. 5.

I find this statement to be misleading. The dimensions do not become more independent as there is a straightforward mathematical dependency between bias and accuracy.

Papers based on the individual differences approach of direct relevance to this ms (more are available):

Jackson, S. A., Kleitman, S., Stankov, L. & Howie, P. (2017). Individual differences in decision making depend on cognitive abilities, monitoring and control. Journal of Behavioral Decision Making, 30(2), 209-223.

THIS PAPER TALKS ABOUT THE CRUCIAL LINK BETWEEN CONFIDENCE JUDGEMENTS AND ACTION CONTROL.

Kleitman, S., & Stankov, L. (2007). Self-confidence and metacognitive processes. Learning and Individual Differences, 17(2), 161–73.

Stankov, L & Crawford, J. D. (1997). Self-confidence and performance on tests of cognitive abilities. Intelligence, 25(2), 93-109.

BOTH PAPERS ADDRESS MEASUREMENT ISSUES OF CONFIDENCE AND ITS CALIBRATION.

We acknowledge that the statement might be considered misleading, and the text including the suggested papers (references 32 & 33):

lines 108-112:

The association between bias and accuracy is complex [27, 28, 32,33]. A highly biased agent can still be very accurate in his judgements by having a very high or low mean confidence score (reflecting over or underconfidence) but discriminating well between correct and incorrect choices. Conversely, a well-calibrated agent (not over nor underconfident) can be terrible at distinguishing between correct and incorrect choices. 

We also utilize the third reference in the discussion:

lines 906-909:

 Similarities involve not only outcomes but also postulated mechanisms. Individuals with low confidence tend to be less decisive [100], give up easier after failing (as do state-oriented people on the preoccupation dimension of action control), and have difficulty overcoming and navigating through daily tasks [101] (as do state-oriented people on the hesitation dimension of action control)

Method/Results:

Why these sample sizes were selected in each study and for each experimental condition (n=30)? And why more participants were not recruited when some did not show up for the second experimental sessions?

Our study focuses on finding at least medium-size effects. This is explained and justified in the paragraph at lines 961-969, showing we have 80% power to detect effects of Cohen’s d = 0.64 or larger, for any given effect. To justify it even further, we performed an additional simulation-based analysis (Supplementary Materials section Power simulation analysis) and modified the power explanation section in the main text:

lines 961-969:

Our tasks have 80% power to detect effects of Cohen’s d = 0.64 or larger. In practice, our model-derived measures can likely reliably detect slightly smaller effect sizes as well, since Hierarchical Bayesian approach is associated with increased efficiency in detecting group differences [114], as are extreme group designs [115]. To account for this, we perform an additional simulation-based analysis (see: Supplementary Materials) and show that we have over 80% power to detect a between-group difference in any given task or condition if the variable of interest is correlated with the ACS score variable at r >= 0.3. Given that our measures of interest were tested across different conditions and experiments, the power to detect such difference in at least one instance is considerably greater. We believe that our null findings in primary task performance across three tasks, two samples of participants and a multitude of performance measures give reasonable support that if such effects exist, they are small and relatively insignificant compared to the differences in confidence. 

We did not recruit additional participants due to limited time to use the experimental facilities.

In results, the authors should report descriptive statistics for ALL of the metrics used, including the estimates of their internal consistency whenever appropriate.

(1) As requested, we added summary tables of descriptive statistics for all directly observable dependent variables (accuracies, RTs and confidence ratings) in the Supplementary Materials (Tables S1-S2).

(2) We added the correlations between ACS subscales and how they relate to the original scale as well as the Polish adaptation

:

lines 418-419:

The correction between the two subscales was r = 0.49, similar to the ACS-90 correlation of r = 0.43 [4] as well as the Polish adaptation (r = 0.45) [50].

Also, was the accuracy of performance analysed in a separate analysis? Were these results presented?

Performance accuracy was analysed for each of the experiments and presented in the results sections. These results are presented in joint sections together with RTs (RT and accuracy). As mentioned in these sections, we perform ANOVAs for each of these dependent measures separately, followed by the description of the results; both the descriptive (means), and inferential. The sections are in the following lines:

RDM Task: lines 428-437

FHD Task: lines 489-505

Experiment 2: 775-790

An optional suggestion: I would also like to see correlations between different metrics, possibly in appendices/sup materials. I’m painfully aware of small sample sizes used in both studies; but still, these estimates might provide further clarifications of what is going on.

Also, was there an effect of gender?

As requested, correlations between measures used in all the tasks have been added in the Supplementary Materials (section Correlations between variables of interest). 

Effects of gender were outside the scope of our interests. Finding out that gender in particular, or any other variable from a limitless list of potential confounders (age, lifestyle, handedness, sexual orientation, etc..), is a significant predictor of any of the dependent measures would not be meaningful from the perspective of our research question or hypotheses.

Discussion in Experiment 1:

The authors state “Confirmation bias assumes biased post-decisional sampling, where evidence confirming choice is associated with a greater gain [78]. This account predicts that the longer information is processed, the stronger the bias should be” (p. 25, lines 595-97). I do not agree with this stipulation, as longer information processing can also indicate that the individual is considering other options, activating system 2, thus resulting in smaller confirmation bias.

 While the mentioned alternative is also possible (longer processing resulting in lower bias) this was not what we hypothesized, nor would that be able to explain the confidence gap effect. This hypothesis was derived from the conceptual framework of action control (Dibbelt & Kuhl, pp 188-190), as well as recent literature showing consistent effects of confirmation bias on evidence processing in similar tasks, and proposing neural implementation mechanisms (Haefner, Berkes & Fiser, 2016; Talluri et al, 2018; Talluri et al, 2021). 

Current statement illustrates a possible mechanistic explanation for the observed differences in confidence magnitudes (along with biased readout), a hypothesis to be later verified in the experiment. 

To make this more clear, we use the term theoretical account

lines 626-627

 This theoretical account predicts that the longer information is processed, the stronger the bias should be, as evidenced by recent findings [79]

The authors also state that “Additionally, some of the results might have been confounded by the ceiling effect and an ambiguous confidence scale” (p. 25, lines 601-2). These two points are unclear. Providing descriptive statistics can help to clarify the existence of ceiling effect. I’m not clear, however, why the authors claim that that the confidence scale used in Experiment 1 was ambiguous? And how was it different in Experiment 2?

(1) Regarding the ceiling effect, we added a reference to the supplementary materials:

lines 593-594

First, the lack of overconfidence might be partially due to the ceiling effect, since the FHD task was performed at a very high level (Supplementary Materials, Table S1)

(2) Ambiguity is explained in 

lines 596-605:

Second, there was an ambiguity associated with the lowest confidence value: 50%. In theory, it should reflect the lowest possible level confidence; that is, when participants make a random guess. In practice, however, it sometimes meant high confidence that the choice was incorrect. This could happen in trials where participants made a motor execution error or noticed an important detail during motor execution when they no longer could alter their choice. This is supported by the fact that the 50% confidence level was chosen almost twice as often than 60% (1644 vs. 985 times) in the FHD task.’

In Experiment 2, this ambiguity was removed by adding an additional confidence level for ‘choice reversals’, i.e., confidence lower than 50% 

lines 730-733:

The linear scale had 7 tick marks, under which the % confidence value was displayed. Values were mapped to keyboard keys from 0 to 6, where 0 represented below 50% confidence (choice reversal) and 6 100%. 

Experiment 2

The authors state:

“Perceptual choices are associated with an objective, external criterion that defines the quality of the choice. Value-based choices are driven by an internally defined criterion supported by subjective evidence. Previous studies comparing the domains suggest similar cognitive mechanisms associated with the decision process [82] and some similarities between their neural signatures [83]. However, we are unaware of any studies comparing metacognitive processing between the two”. P. 26, lines 618-623.

This statement is incorrect as there is extensive previous research available. E.g., (and there are more):

Kleitman, S., & Stankov, L. (2001). Ecological and person-driven aspects of metacognitive processes in test-taking. Applied Cognitive Psychology, 15, 321–41. doi:10.1002/acp.705

Stankov, L. (1998). Calibration curves, scatterplots and the distinction between general knowledge and perceptual tasks. Learning and Individual Differences, 10(1), 29-50.

While this study (and many others) compares confidence across different task domains, we refer specifically to perceptual and value-based domains. This is crucial, since value-based choices are inherently subjective, as opposed to perceptual, memory, or knowledge-based tasks. To make this point more explicit, we modified the sentence at:

lines 652-653

However, we are unaware of any studies comparing metacognitive processing between these two domains.

Method/Results:

Please justify sample sizes and provide clarifications specified for Experiment 1 (e.g., table with descriptive statistics etc., calibration curves). Were there gender differences?

(1) Sample size discussion have been modified in the main text (see one of the previous comments relating to this question). To further justify the sample sizes, we added an additional simulation-based power analysis in the Supplementary Materials (section Power simulation analysis). 

(2) We added a descriptive statistics table to the Supplementary Materials (Tables S1-S2)

(3) We added calibration plots to the Supplementary Materials (section Confidence calibration curves).

Discussion/general discussion

“Our results expand on the literature providing a universal mechanism shaping basic decision-making even in the absence of affective manipulation.” (p. 34). I find the use of the word “universal” to be a bit inflated, especially given the limitations described above. Also, what seems to be missing for me is a discussion of how despite using completely different method and analytics, some of these results extend and replicate the results obtained via using the individual differences paradigm.

Thank you for pointing this out, we replaced the term ‘universal’ with the more concrete ‘across-domain’ (line 876), as the experiment proves the differences to hold in two separate domains (value-based and preference-based choice tasks).

Reviewer #1a

1. There seems to be considerable differences between the theories surrounding High and Low Action Control individuals and the results of the current experiment. It appears that much of the Introduction is on how action-oriented individuals have beneficial attributes, particularly those surrounding flexible decision-making in more challenging situations whilst state-oriented individuals cannot adjust plans and will fail in their decision-making. From the results, it appears that the only difference between the two groups is that one is more confident than the other. The authors mention that without a time pressure manipulation, it was hard to induce differences. However, I don’t see how this is a limitation considering participants had generally responded to items within 1 second.

Across the study, we think of time pressure as a relative, not an absolute measure. Having 2 seconds to respond in an easy or unimportant task can be more than enough time. Conversely, having an hour to make a difficult life-changing decision can be considered a high time-pressure situation. In the RDM task, time pressure was manipulated by task instructions. Participants were instructed to focus either on speed (high time pressure) or accuracy (low time pressure) in their response. This manipulation was successful, as reaction times were in fact longer in the accuracy focus condition. Of course, it is impossible to say how much time pressure participants experienced in the accuracy condition but considering the fact that most of them responded on average much faster than the time limit in the accuracy focus condition (mean of 750ms), might suggest that they considered to have enough time. Hence, when mentioning our results pertaining to time pressure, we reference this particular experimental manipulation.

2. The action control variable appears to be similar in definition to adaptability and resilience, specifically within the decision-making domain. It would be good to mention this to some extent.

While this might be true, to our knowledge no studies directly related adaptability or resilience to individual differences in action-control, and the definitions of these concepts can be rather broad. In fact, there is little consensus on how resilience should be defined (de Terte & Stephens, 2014). Based on this, we find little value in relating action-control to these concepts, as opposed to providing the definition, experimental findings, and experimentally confirmed links with other psychological constructs. More informed readers are welcomed to make connections with concepts of their theoretical interest, this is however not a goal of the current paper.

3. Although I understand the benefits of confidence, particularly those stated in the conclusion, it is important to point out the differences in metacognitive accuracy, or lack thereof. The current study indicates that there is either a greater or similar metacognitive accuracy comparing action-oriented and state-oriented groups. In this sense, it’s important to note that action-oriented individuals appear to have greater confidence without the negative aspect of overconfidence.

This is indeed important. We addressed this point in the general discussion.

lines 859-865

Here, by rigorously testing choice and post-choice cognitive processing, we show that this trait is systematically related to post-decisional metacognitive processing. Action-oriented people display higher levels of choice confidence across task domains, choice difficulty, time pressure, and deliberation period. This confidence gap is dissociated from measures of performance which show no meaningful differences, indicating that higher confidence displayed by action-oriented people does not necessarily carry the negative burden of being less accurate in their assessments.

Minor Comments:

- In the introduction, the authors mention that Action Control is related to differences in cognitive control and ability citing moderate positive correlations with consciousness and self-monitoring. Although I understand the self-monitoring relationship with cognitive control, I don’t see any mention of the relationships there with cognitive ability (i.e., intelligence).

Thank you for pointing this out, the term ‘ability’ is in fact an incorrect and have been removed (line 59).

- In the 2nd experiment, are participants allowed to rate preferences similarly across the items (e.g. a preference for 80 for both)? If this is the case, how would accuracy be computed if the two are paired with each other?

This indeed was not clarified in the initial text. Clarification now added:

lines 734-739:

Choice pairs in the main task were constructed using an algorithm which 1) calculated the value difference (distance) between all available item pairs (n = 2485), 2) created easy and difficult choice groups by taking pairs where distance is greater than half of the maximum (easy) or smaller than 1/8 of the maximum, excluding pairs that are equal (difficult), 3) sampled 40 pairs randomly out of each of the 2 groups.

- Please label the y axes of the Experiment 2 figures."

Y-axis labels are provided above the plots. We added (%) following Confidence Magnitude to make the connection more clear.

Reviewer #2

The paper presents a well-designed, novel, and interesting study with results that could potentially make an important contribution to the literature. The drawbacks of the study include small sample size which may have limited the possibility to detect between-subject effects and their interactions, as well as complexity of the paper, which makes it difficult to understand and appreciate the findings. As a general recommendation to improve the paper readability, I would suggest to make more connection in the discussion to the original hypotheses, and more comparison and connection of the findings of the two experiments. I would like to point out some additional minor points in the text where additional detail could benefit the presentation.

We address the readability issues by providing more explicit links between hypotheses and the following discussion sections for both experiments. More specifically, we:

- add a table describing the model-derived parameters of interest in Experiment 1 in more detail (Table 1, lines 179-182)

- add references to the numbered hypotheses in Experiment 1 discussion (lines 574-576)

- add numbered hypotheses to Experiment 2 (lines 686-698)

- link these hypotheses directly to the confidence gap, which is the main finding of Experiment 1 (lines 684-685)

- add references to the numbered hypotheses in Experiment 2 discussion (lines 827, 829, 833)

Sample size

Our study focuses on finding at least medium-size effects. This is explained and justified in general discussion (lines 959-972), showing we have 80% power to detect effects of Cohen’s d = 0.64 or larger, for any given effect. To justify it even further, we performed an additional simulation-based analysis (Supplementary Materials, section Power simulation analysis) which focuses on the benefits of the extreme-group design. We modified the power explanation section in the main text accordingly:

lines 961-972:

Our tasks have 80% power to detect effects of Cohen’s d = 0.64 or larger. In practice, our model-derived measures can likely reliably detect slightly smaller effect sizes as well, since Hierarchical Bayesian approach is associated with increased efficiency in detecting group differences [113], as are extreme group designs [114]. To account for this, we perform an additional simulation-based analysis (see: Supplementary Materials) and show that we have over 80% power to detect a between-group difference in any given task or condition if the variable of interest is correlated with the ACS score variable at r >= 0.3. Given that our measures of interest were tested across different conditions and experiments, the power to detect such difference in at least one instance is considerably greater. We believe that our null findings in primary task performance across three tasks, two samples of participants and a multitude of performance measures give reasonable support that if such effects exist, they are small and relatively insignificant compared to the differences in confidence. 

Line 104: “The association between bias and accuracy is complex. Both extremely high and low bias would lead to poor accuracy” : perhaps, saying “strong positive” and “strong negative” instead of “high” and “low” would be clearer.

As this is something pointed out by reviewer 1 as well, we adjusted the following 

lines: 108-112

The association between bias and accuracy is complex [27,28,32,33]. A highly biased agent can still be very accurate in his judgements by having a very high or low mean confidence score (reflecting over or underconfidence) but discriminating well between correct and incorrect choices. Conversely, a well-calibrated agent (not over nor underconfident) can be terrible at distinguishing between correct and incorrect choices. 

Line 110: Adding a few words to clarify the notion of “alternative spreading” for readers familiar with the concept, but not the term, would be good (e.g., “alternative spreading (opposite changes in the attractiveness of the preferred and the competing alternatives)”)

The line has been adjusted to:

lines 117-119

Indeed, state-orientation has been shown to reduce alternative spreading, a phenomenon where alternative value grows after being chosen, and shrinks after being rejected [34] - one of the hallmarks of cognitive dissonance reduction [35]. 

Line 258-260: It might be good to quantify the magnitude of the difference between the contrast groups (as Cohen’s d) here and in Experiment 2, to make it clear how huge the effect size is, thanks to the large initial sample.

Line 280: It would be better to report the correlations between the ACS subscales in the present sample (N = 724), either here or below (Line 396).

(1) We added the requested details in both sample descriptions using – as a reference – the distribution of scores obtained in our screening sample.

lines 275-277 (Experiment 1):

 When compared against the distribution of all ACS scores obtained in the screening phase, the means in two contrast groups were separated by 2.81 SD.

lines 705-707 (Experiment 2):

 Standardized between-group difference in mean assessed against the distribution of ACS scores from the screening sample was equal to 2.64.

(2) Correlation analysis has been added:

 lines 418-419:

The correction between the two subscales was r = 0.49, similar to the ACS-90 correlation of r = 0.43 [4] as well as the Polish adaptation (r = 0.45) [50].

Line 310: From the description of Procedure it is not clear whether the study was carried out in a lab setting or online.

We added the following sentence to make this clear

line 340-341:

All tasks were performed offline took place in a testing lab on the premises of the university. 

Line 355: “Conventionally, differences larger than 10 in DIC scores are considered significant [61]” : this appears to simplify things a bit too much, given that Spiegelhalter et al. (2002, p. 613) write: “Burnham and Anderson (1998) suggested models receiving AIC within 1–2 of the ‘best’ deserve consideration, and 3–7 have considerably less support: these rules of thumb appear to work reasonably well for DIC.” Based on this, the authors’ approach to interpreting DIC appears to be an oversimplification with ramifications for model choice: “All the models displayed very similar DIC scores (between 8441 and 8448), indicating that adding action control as a between-person factor did not significantly improve model fit.” (Line 445) : it looks like the models with a DIC difference of 7 can hardly be deemed equivalent (although several models with DIC 8441-8443 can be?)? Because this rule of thumb might have substantive consequences, a more careful treatment of DIC seems necessary. I would suggest not to interpret the differences in DIC between models as “significant”, but rather in terms of relatively better/worse model fit.

This is a fair point. While the ‘10 point rule’ is far from an objective standard, it has been used in many previous studies that use the HDDM toolbox. This rather conservative threshold is driven by the fact that DIC is an imperfect, biased measure (Wiecki, Sofer & Frank, 2013) and can be a subject to noise (this is based on personal experience with HDDM, where the DIC estimations can sometimes vary up to 2-3 points). While using a non-threshold approach is perfectly valid, the threshold-based analysis was our a priori method of choice and we think it conveys a clear message, while not impacting the inference in a meaningful way. In order to explain this better, we change the following

lines 371-377

We compare the models using Deviance Information Criterion (DIC) [58], a metric of fit that takes into account model complexity, where lower scores indicate better fit. Since DIC can be a biased and imperfect measure [55], we adopt a conservative, proven as a reliable threshold in other works that adopt the HDDM toolbox. Conventionally, differences larger than 10 in DIC scores are considered significant [59]. While this assumption might be considered an oversimplification, it has proven to work well and has been adapted in many empirical works using the HDDM toolbox [60, 61, 62]

Line 423: “Condition-dependent” probably needs a dash.

Thank you for pointing this out. It has been corrected

Lines 420-430: A description of DDM parameters might be better placed to the Methods section.

We believe that an early introduction to the basic DDM parameters in the introduction is quite important for a full understanding of the tested hypotheses. We added a reference to this in the methods:

lines 349-351

These include DDM parameters (threshold, threshold difference, starting point bias and drift rate; see Experiment 1 introduction),

Lines 542-543: The legend for Figure 2 needs to explain all the abbreviations (ACC, SP, CON, NEU, INC).

Legend has been adjusted.

lines 566-570:

Fig 2. Significant group differences in confidence magnitude and accuracy across the two tasks (RDM: random dot motion; FHD: Face/house discrimination). ACC and SP refer to the accuracy emphasis and speed emphasis conditions in the RDM task, respectively. CON, NEU and INC refer to the congruent, neutral and incongruent conditions in the FHD task, respectively. Error bars represent standard errors across participants.

Line 546: It would be good if the results and discussion of Experiment 1 made reference to the numbered hypotheses presented at the beginning.

We address this comment in the beginning of the Experiment one discussion section

lines 573-581:

Experiment 1 results paint a clear picture: confidence was the only measure consistently differentiating the groups, supporting the claim of differential post-decisional maintenance between groups (hypothesis 4). We found no support for hypotheses 1 to 3, postulating differences in the decision-making process.

Line 588: “These results align with action-driven theories of post-decisional processing.” An explanatory sentence describing the predictions of these theories would be good here.

We added a brief clarification

lines 617-619

These results align with action-driven theories of post-decisional processing, which postulate its central role in facilitating action [77].

Line 654: It would be great to present the study goals in a more structured way of numbered hypotheses, as in the description of the first experiment.

As suggested, we add structured hypotheses in Experiment 2 introduction

lines 684-698

Based on these study goals, we derive the following hypotheses with regards to the differences in confidence between groups (confidence gap) for Experiment 2.

Hypothesis 1. The confidence gap is domain-general. We expect to observe a between-group difference in confidence in both perceptual and value-based tasks.Hypothesis 2a. The confidence gap is driven by biased sampling of evidence after the choice is made. This account predicts the gap to widen as a function of time given for consideration, after the choice is made.

Hypothesis 2b. The confidence gap is driven by a biased read-out, manifested in biased sampling of evidence after the choice is made. This account predicts the gap to exist irrespective of consideration time.

Hypothesis 3. The confidence gap is affected by choice difficulty. We expect the gap to widen in the difficult choice condition.

Note that hypotheses 2a and 2b are not exclusive. If both accounts are true, we would expect to observe a significant difference in confidence in time point one (immediately after the choice) which then gets wider at time point two. 

Line 672: How exactly was the task formulated for participants? The text mentions “size” and “surface area”, which might not be completely interchangeable.

This has now been clarified.

lines 717-722:

In perceptual trials, participants decided which of the items has is larger, while in the value-based they chose the item they would prefer to win. The instruction clarified that in the perceptual condition, size refers to the item surface area, and can be quantified as the percentage of non-white (item-related) pixels to white background pixels (so that an empty white image would have a size of 0, and an image where the item takes all the pixels would have a size of 100).

Line 685-686: The preference ratings are based on each object being rated by a participant on a 100-point scale and just once. The following procedure using these individual ratings to generate easy/difficult pairs treats preference ratings as perfectly measured. This approach is totally fine for size, which is an objective property, but for value one can expect some measurement error (which might be fairly large, given the single-item nature of the measure). For “difficult” pairs, the distance between two items being compared might actually be smaller than the uncertainty associated with the error of measurement (which, unfortunately, is unknown). As a result, an item rated 50% might really be less preferred than one rated 48%. The untenable assumption of perfect measurement makes it hardly possible to interpret accuracy ratings for values (at least for difficult choices), which needs to be reflected in the presentation of the results and discussion.

This is indeed true. We added a paragraph explaining this in the methods section.

lines 740-748:

Accuracy is operationalized as either choosing the larger (perceptual condition) or more preferred item (value-based condition). A limitation of accuracy analysis in the value-based condition is the assumption that the initial value ratings are noiseless, which is unlikely to be true. In reality, an item rated at 60% might be less preferred than one rated at 58%. Due to this, accuracy might be a slightly inaccurate measure, especially in difficult trials. We do not expect this confound to meaningfully affect inference, since we have no reason to believe that the two groups differ in terms of the initial noisiness of item evaluation. To account for this possibility however, we also compare the mean and spread of initial ratings between the groups.

(As an additional possibility, it appears that individuals did not have a time limit and could change their answers, which may have introduced some metacognitive bias, especially for state-oriented individuals. Could it be that the size and the distribution of measurement error in deriving initial value ratings was different in action and state-oriented individuals?)

The initial distributions have been compared in the Item value ratings paragraph of the results section (lines 762-766), where we find no differences between the means and standard deviations between groups.

References

Boldt, A., Schiffer, AM., Waszak, F. et al. Confidence Predictions Affect Performance Confidence and Neural Preparation in Perceptual Decision Making. Sci Rep 9, 4031 (2019).

de Terte, Ian; Stephens, Christine (2014). Psychological Resilience of Workers in High-Risk Occupations. Stress and Health, 30(5), 353–355. doi:10.1002/smi.2627 

Elman, J.A., Klostermann, E.C., Marian, D.E. et al. Neural correlates of metacognitive monitoring during episodic and semantic retrieval. Cogn Affect Behav Neurosci 12, 599–609 (2012). 

Fleming SM, Massoni S, Gajdos T, Vergnaud J-C. Metacognition about the past and future: quantifying common and distinct influences on prospective and retrospective judgments of self-performance. Neurosci Conscious . 2016 Jan 1;2016(1). 

Navajas J, Bahrami B, Latham PE. Post-decisional accounts of biases in confidence. Curr Opin Behav Sci. 2016;11:55–60.

Haefner RM, Berkes P, Fiser J. Perceptual Decision-Making as Probabilistic Inference by Neural Sampling. Neuron. 2016 May 4;90(3):649-60. doi: 10.1016/j.neuron.2016.03.020. Epub 2016 Apr 14. PMID: 27146267.

Talluri BC, Urai AE, Tsetsos K, Usher M, Donner TH. Confirmation Bias through Selective Overweighting of Choice-Consistent Evidence. Curr Biol. 2018 Oct 8;28(19):3128-3135.e8. doi: 10.1016/j.cub.2018.07.052. Epub 2018 Sep 13. PMID: 30220502.

Talluri BC, Urai AE, Bronfman ZZ, Brezis N, Tsetsos K, Usher M, Donner TH. Choices change the temporal weighting of decision evidence. J Neurophysiol. 2021 Apr 1;125(4):1468-1481. doi: 10.1152/jn.00462.2020. Epub 2021 Mar 10. PMID: 33689508.

Wiecki T V, Sofer I, Frank MJ. HDDM: Hierarchical Bayesian estimation of the Drift-Diffusion Model in Python. Front Neuroinfor. 2013 Aug 2;7:14.

---

## [Decision Letter · Decision Letter 1]

18 Feb 2022

PONE-D-21-08854R1Are you confident enough to act? Individual differences in Action Control are associated with post-decisional metacognitive biasPLOS ONE

Dear Dr. Zajkowski,

Thank you for submitting your manuscript to PLOS ONE. After careful consideration, we feel that it has merit but does not fully meet PLOS ONE’s publication criteria as it currently stands. Therefore, we invite you to submit a revised version of the manuscript that addresses the points raised during the review process.

We look forward to receiving your revised manuscript.

Kind regards,

Dragan Pamucar

Academic Editor

PLOS ONE

Journal Requirements:

Reviewers' comments:

Reviewer's Responses to Questions

**Comments to the Author**

1. If the authors have adequately addressed your comments raised in a previous round of review and you feel that this manuscript is now acceptable for publication, you may indicate that here to bypass the “Comments to the Author” section, enter your conflict of interest statement in the “Confidential to Editor” section, and submit your "Accept" recommendation.

Reviewer #1: (No Response)

Reviewer #2: All comments have been addressed

2. Is the manuscript technically sound, and do the data support the conclusions?

Reviewer #1: Partly

Reviewer #2: Yes

3. Has the statistical analysis been performed appropriately and rigorously? 

Reviewer #1: No

Reviewer #2: Yes

4. Have the authors made all data underlying the findings in their manuscript fully available?

Reviewer #1: Yes

Reviewer #2: Yes

5. Is the manuscript presented in an intelligible fashion and written in standard English?

Reviewer #1: Yes

Reviewer #2: Yes

6. Review Comments to the Author

Reviewer #1: Dear Authors, I believe this manuscript version is improved, but I still have several questions and concerns.

1) I was pleased to see calibration curves and correlations added in the appendices. However, no interpretations have been offered. Please provide them, especially about your findings, e.g., how accuracy and confidence, confidence and meta-d' and accuracy and meta-d' correlations may have affected your results and their interpretations (when they are high and then they are low).

2) Is it possible to adjust the formatting of your correlation tables. I'm aware that R produced them, but I wonder whether you can fix the font and appearance.

3) I question your main conclusion that "We propose that a high confidence bias might be crucial for the successful realization of intentions in many real-life situations". A high confidence bias is linked to arrogance and not such great life outcomes (see

Bruine de Bruin, W., Missier, F. D., & Levin, P. I. (2012). Individual differences in decision-making competence. Journal of Behavioral Decision Making, 25, 329–330.

Bruine de Bruin, W., Parker, A. M., & Fischhoff, B. (2007). Individual differences in adult decision-making competence. Journal of Personality and Social Psychology, 92(5), 938.

Kleitman, S., Hui, J.SW. & Jiang, Y. (2019). Confidence to spare: individual differences in cognitive and metacognitive arrogance and competence. Metacognition Learning, 14(3), 479–508 https://doi.org/10.1007/s11409-019-09210-x).

Dunning, D., Johnson, K., Ehrlinger, J., & Kruger, J. (2003). Why people fail to recognize their own incompetence. Current Directions in Psychological Science, 12(3), 83–87.

Del Missier, F., Mäntylä, T., & Bruin, W. B. (2012). Decision-making competence, executive functioning, and general cognitive abilities. Journal of Behavioral Decision Making, 25(4), 331–351.

Please compare and contrast your research results with these studies.

The significant limitations of your studies are small sample sizes, a limited selection of your research task, some stats dependency of your metrics, narrow range. Given all these limitations, I believe you should 'soften' your conclusions and consider possible alternative explanations.

4) the paper is very dense. I think it would benefit from good editing/proofreading.

Reviewer #2: Thank you for addressing the feedback and adding the missing details, as well as supplementary analyses to clarify things. I am completely satisfied with the revision and have no more substantial recommendations, except for one: please edit the newly incorporated text fragments for language and readability, e.g.:

"All tasks were performed offline [and] took place in a testing lab on the premises of the university." (p. 15)

"To account for this, we perform an additional simulation-based analysis (see: Supplementary Materials) and show that we have over 80% power..." (p. 40) -- it is better to explain the steps of the analyses in the past tense ("we perfomed... and found...") and only the conclusions in the present tense.

"We found no support for hypotheses 1 to 3, postulating differences in the decision-making process." (p. 25) -- the comma is probably not needed, as it is a reduced relative clause.

There are also similar minor issues in other fragments.

7. PLOS authors have the option to publish the peer review history of their article (what does this mean?). If published, this will include your full peer review and any attached files.

Reviewer #1: No

Reviewer #2: No

---

## [Author Response · Author response to Decision Letter 1]

5 Apr 2022

Reviewer 1.

Dear Authors, I believe this manuscript version is improved, but I still have several questions and concerns.

1) I was pleased to see calibration curves and correlations added in the appendices. However, no interpretations have been offered. Please provide them, especially about your findings, e.g., how accuracy and confidence, confidence and meta-d' and accuracy and meta-d' correlations may have affected your results and their interpretations (when they are high and then they are low).

We are thankful for the constructive remarks. In order to address these issues, we added the following paragraphs in the Supplementary Materials:

Regarding the calibration curve figures:

‘The calibration patterns are dependent on task type and difficulty. Both groups display characteristic underconfidence in difficult conditions (Figures S1, S3), as well as slight overconfidence when the task is easy (Figures S2, S3 ,S4) [76]. The main between-group difference is the effect of the state-oriented group being overall more under confident when the task is easy, and less overconfident when the task is difficult. The three plots also speak against simpler heuristic explanations, such as the difference between-groups being driven by the propensity of either action-oriented participants using the highest confidence rating, or state-oriented participants using the lowest.’

Regarding correlation figures:

‘The correlation tables indicate that the relations between variables of interest vary depending on task and condition. The relation between accuracy and confidence is significant only for the FHD task, i.e. the task with the highest level of accuracy. This is not surprising, since the easier the task, the easier it is to assess one’s own performance, and overall (assuming bias averages out) we should expect more accurate actors to display more confidence.

Similar pattern can be observed between confidence and meta-d’ measures, where significant correlations can also be found in the easier experimental conditions, suggesting more confident participants were also more metacognitively accurate. Meta-d’ measure also covaries with accuracy in a number of experimental conditions, suggesting type 1 and type 2 performance being associated on a between-person level (i.e., more accurate participants tended to also be better at metacognitive discrimination). ‘

2) Is it possible to adjust the formatting of your correlation tables. I'm aware that R produced them, but I wonder whether you can fix the font and appearance.

To accommodate this request, we changed the correlation plots to standard tables.

3) I question your main conclusion that "We propose that a high confidence bias might be crucial for the successful realization of intentions in many real-life situations". A high confidence bias is linked to arrogance and not such great life outcomes (see

Bruine de Bruin, W., Missier, F. D., & Levin, P. I. (2012). Individual differences in decision-making competence. Journal of Behavioral Decision Making, 25, 329–330.

Bruine de Bruin, W., Parker, A. M., & Fischhoff, B. (2007). Individual differences in adult decision-making competence. Journal of Personality and Social Psychology, 92(5), 938.

Kleitman, S., Hui, J.SW. & Jiang, Y. (2019). Confidence to spare: individual differences in cognitive and metacognitive arrogance and competence. Metacognition Learning, 14(3), 479–508 http://dx.doi.org/10.1007/s11409-019-09210-x).

Dunning, D., Johnson, K., Ehrlinger, J., & Kruger, J. (2003). Why people fail to recognize their own incompetence. Current Directions in Psychological Science, 12(3), 83–87.

Del Missier, F., Mäntylä, T., & Bruin, W. B. (2012). Decision-making competence, executive functioning, and general cognitive abilities. Journal of Behavioral Decision Making, 25(4), 331–351.

Please compare and contrast your research results with these studies.

The significant limitations of your studies are small sample sizes, a limited selection of your research task, some stats dependency of your metrics, narrow range. Given all these limitations, I believe you should 'soften' your conclusions and consider possible alternative explanations.

In order to explore the relation between our findings and the research on overconfidence, we added a paragraph in the general discussion

Lines 903 - 917:

‘It is also important to contrast our findings concerning the confidence gap (and its known positive correlates) with results underlining detrimental effects of excessive confidence, such as those related to arrogance [102] or political radicalization [103]. Extreme levels of overconficence might also lead to faulty assessments, unrealistic expectations and hazardous decisions [104], which suggests the possibility that the relationship between confidence and beneficial real-life outcomes is not monotonous. At the same time, it also might be worth noting that studies focusing on confidence calibration usually do not distinguish between bias and sensitivity [105-107], making the measure susceptible to being influenced by both factors. Detrimental overconfidence requires both – a high bias, i.e. a tendency to display high levels of confidence in general, as well as poor sensitivity, i.e. the inability to change one’s mind in light of contradicting evidence. 

In addition, in our study, both groups displayed characteristic patterns of overconfidence when the task was easy and underconfidence when it was difficult [76] (Supplementary Materials) highlighting that the difference was not qualitative, but quantitative in nature. ‘

Since the distinction between bias and overconfidence is not obvious we also clarified the sentence mentioned in the abstract, to make it less ambiguous:

Lines 44 - 46:

‘We propose that a positive confidence bias, coupled with appropriate metacognitive sensitivity, might be crucial for the successful realization of intentions in many real-life situations.’

In addition, we added a paragraph in the limitations section, addressing some alternative explanations:

Lines 979 - 986:

‘Fourthly, given the limited scope of the study (limited sample, set of experimental tasks and measures used), we cannot fully rule-out some alternative explanations. E.g. performance in a controlled laboratory environment might not be ecologically valid and does not always correspond to real-life situations [121]. Another factor is that confidence ratings are based on self-report scale, making it susceptible to additional confounds. In this light, state-oriented individuals might only express lower confidence while not truly experiencing it. Such report bias [122] could be driven by factors including modesty or excessive self-monitoring [123].’

We also modified the Conclusions section by softening the sentence in the first paragraph:

Lines 1000 - 1002:

‘Our results suggest that the differences between groups arise not due to how evidence is obtained or strategic adjustments but due to how it is interpreted by a higher-level process relating evidence to subjective confidence.’

We also added a sentence explicitly stating the necessity of further studies in order to generalize the findings beyond the realm of value-based and perceptual tasks:

Lines 1014 - 1017:

‘Further research using different tasks and measures is necessary to confirm these speculations and extrapolate the generalizability of our findings to domains outside of perception and value-based decision tasks.’

4) the paper is very dense. I think it would benefit from good editing/proofreading.

Acknowledging this, we have proof-read the paper, correcting many minor issues and making it clearer.

Reviewer #2 

Thank you for addressing the feedback and adding the missing details, as well as supplementary analyses to clarify things. I am completely satisfied with the revision and have no more substantial recommendations, except for one: please edit the newly incorporated text fragments for language and readability, e.g.:

"All tasks were performed offline [and] took place in a testing lab on the premises of the university." (p. 15)

"To account for this, we perform an additional simulation-based analysis (see: Supplementary Materials) and show that we have over 80% power..." (p. 40) -- it is better to explain the steps of the analyses in the past tense ("we perfomed... and found...") and only the conclusions in the present tense.

"We found no support for hypotheses 1 to 3, postulating differences in the decision-making process." (p. 25) -- the comma is probably not needed, as it is a reduced relative clause.

There are also similar minor issues in other fragments.

Thank you for pointing out these language issues. We now corrected these, as well as many other ones throughout the text (corrections visible in the Revised Manuscript with Track Changes)

---

## [Decision Letter · Decision Letter 2]

3 May 2022

Are you confident enough to act? Individual differences in Action Control are associated with post-decisional metacognitive bias

PONE-D-21-08854R2

Dear Dr. Zajkowski,

We’re pleased to inform you that your manuscript has been judged scientifically suitable for publication and will be formally accepted for publication once it meets all outstanding technical requirements.

Kind regards,

Dragan Pamucar

Academic Editor

PLOS ONE

Additional Editor Comments (optional):

Reviewers' comments:

Reviewer's Responses to Questions

**Comments to the Author**

1. If the authors have adequately addressed your comments raised in a previous round of review and you feel that this manuscript is now acceptable for publication, you may indicate that here to bypass the “Comments to the Author” section, enter your conflict of interest statement in the “Confidential to Editor” section, and submit your "Accept" recommendation.

Reviewer #2: All comments have been addressed

2. Is the manuscript technically sound, and do the data support the conclusions?

Reviewer #2: Yes

3. Has the statistical analysis been performed appropriately and rigorously? 

Reviewer #2: Yes

4. Have the authors made all data underlying the findings in their manuscript fully available?

Reviewer #2: Yes

5. Is the manuscript presented in an intelligible fashion and written in standard English?

Reviewer #2: Yes

6. Review Comments to the Author

Reviewer #2: Thank you for revising the manuscript, in particular, with respect to the language. I have no further comments.

7. PLOS authors have the option to publish the peer review history of their article (what does this mean?). If published, this will include your full peer review and any attached files.

Reviewer #2: No

---

## [Editor Report · Acceptance letter]

9 May 2022

PONE-D-21-08854R2 

Are you confident enough to act? Individual differences in Action Control are associated with post-decisional metacognitive bias 

Dear Dr. Zajkowski:

I'm pleased to inform you that your manuscript has been deemed suitable for publication in PLOS ONE. Congratulations! Your manuscript is now with our production department. 

Kind regards, 

on behalf of

Dr. Dragan Pamucar 

Academic Editor

PLOS ONE